# A New Exopolysaccharide of Marine Coral-Associated *Aspergillus pseudoglaucus* SCAU265: Structural Characterization and Immunomodulatory Activity

**DOI:** 10.3390/jof9111057

**Published:** 2023-10-27

**Authors:** Bo Peng, Yongchun Liu, Yuqi Lin, Supaluck Kraithong, Li Mo, Ziqing Gao, Riming Huang, Xiaoyong Zhang

**Affiliations:** 1Institute for Environmental and Climate Research, Jinan University, Guangzhou 511443, China; pengbo@jnu.edu.cn; 2University Joint Laboratory of Guangdong Province, Hong Kong and Macao Region on Marine Bioresource Conservation and Exploitation, College of Marine Sciences, South China Agricultural University, Guangzhou 510642, China; yongchunliu626@163.com (Y.L.); molly5792@163.com (L.M.); q18588577456@163.com (Z.G.); 3Guangdong Provincial Key Laboratory of Nutraceuticals and Functional Foods, College of Food Science, South China Agricultural University, Guangzhou 510642, China; 13928491517@163.com (Y.L.); supaluck@scau.edu.cn (S.K.)

**Keywords:** exopolysaccharide, coral-associated *Aspergillus pseudoglaucus* SCAU265, immunomodulatory activity, amino acid metabolism

## Abstract

Recent studies have found that many marine microbial polysaccharides exhibit distinct immune activity. However, there is a relative scarcity of research on the immunomodulatory activity of marine fungal exopolysaccharides. A novel water-soluble fungal exopolysaccharide ASP-1 was isolated from the fermentation broths of marine coral-associated fungus *Aspergillus pseudoglaucus* SCAU265, and purified by Diethylaminoethyl-Sepharose-52 (DEAE-52) Fast Flow and Sephadex G-75. Structural analysis revealed that ASP-1 had an average molecular weight of 36.07 kDa and was mainly composed of (1→4)-linked α-D-glucopyranosyl residues, along with highly branched heteropolysaccharide regions containing 1,4,6-glucopyranosyl, 1,3,4-glucopyranosyl, 1,4,6-galactopyranosyl, T(terminal)-glucopyranosyl, T-mannopyranosyl, and T-galactopyranosyl residues. ASP-1 demonstrated significant effects on the proliferation, nitric oxide levels, and the secretion of cytokines TNF-α and IL-6 in macrophage RAW264.7 cells. Metabolomic analysis provided insights into the potential mechanisms of the immune regulation of ASP-1, suggesting its involvement in regulating immune function by modulating amino acid anabolism, particularly arginine synthesis and metabolism. These findings provide fundamental scientific data for further research on its accurate molecular mechanism of immunomodulatory activity.

## 1. Introduction

Marine invertebrates are recognized as valuable sources of biological compounds owing to their unique ecological habitat [1]. However, recent studies have found that many of these biological compounds are produced by their symbiotic microorganisms [2]. The study of biological compounds from symbiotic microorganisms in marine invertebrates is currently a hot topic of research. In contrast to small molecules derived from marine sources, the characterization and bioactivity of extracellular polysaccharides from marine-derived microorganisms present a greater challenge due to their structural complexity and diversity [3]. Nevertheless, the growing interest in microbial extracellular polysaccharides has revealed their diverse chemical structures and significant bioactivities, making them promising for human health applications due to their low toxicity. For instance, Chen et al. purified and identified an exopolysaccharide (AS2-1) from the marine-derived fungus *Alternaria* sp. SP-32, which exhibited unique chemical properties along with notable antioxidant activity [4]. Similarly, a polysaccharide (HPA) isolated from the marine-derived fungus *Hansfordia sinuosae* HS demonstrated a strong inhibitory effect on human carcinoma MCF-7 and HeLa cells [5]. Moreover, an extracellular polysaccharide (EAPS) derived from *Rhodotorula* sp. RY1801 exhibited a very strong immunomodulatory effect on nematodes [6]. These findings collectively highlight the potential of marine fungi as important sources of biologically active extracellular polysaccharides with substantial application value.

*Aspergillus*, a significant genus within the Ascomycota phylum, exhibits a wide distribution across various environments and has proven instrumental in the production of diverse products. In the marine environments, *Aspergillus* species serve as valuable sources of biological secondary metabolites, with numerous compounds having been isolated from different marine-derived *Aspergillus* species [7]. These compounds encompass diketopiperazine alkaloids and sulfoxide-containing bisabolane sesquiterpenoids [8], antimicrobial and cytotoxic dipeptides [9], polyketides [10], meroterpenoids [11], and indole glucosides [12]. Despite the extensive research on secondary metabolites, there has been limited investigation into biological extracellular polysaccharides derived from marine *Aspergillus* species. Some notable examples include the exopolysaccharide AVP-141 from marine fungus *Aspergillus versicolor* SCAU141 [13], the antioxidant extracellular polysaccharide YSS from *A. terreus* [14], the exopolysaccharide AWP isolated from a marine coral-associated *A. versicolor* [15], and an antioxidant exopolysaccharide isolated from a marine-derived *Aspergillus* sp. Y1 [16]. However, research on the characteristics of the chemical structure and bioactivity of marine polysaccharides from *Aspergillus* species remains limited.

In the present study, our objective was further investigating the potential application of exopolysaccharides from marine-derived *Aspergillus* spp., elucidating their structural properties and biological activities, particularly their immunomodulatory function. We conducted extraction, purification, and characterization of a novel exopolysaccharide ASP-1 derived from the marine coral-associated fungus *Aspergillus pseudoglaucus* SCAU265, focusing on its physicochemical properties and immunomodulatory effects. ASP-1 was obtained through fungal fermentation followed by ethanol precipitation and column chromatography. And the molecular weight, glycosidic bonds, and monosaccharide compositions of ASP-1 were investigated by Fourier transform–infrared spectroscopy (FTIR), gas chromatography–mass spectrometry (GC–MS), and nuclear magnetic resonance (NMR) spectroscopy, etc. Additionally, we evaluated the immunomodulatory activity of ASP-1 on RAW264.7 macrophages by assessing its effects on cell viability, and nitric oxide (NO) release, interleukin-6 (IL-6), and tumor necrosis factor-alpha (TNF-α) secretion. Furthermore, we conducted metabolomics analysis to uncover the potential molecular mechanism underlying the immunomodulatory effects of ASP-1.

## 2. Materials and Methods

### 2.1. Fungal Strain and Fermentation

The fungal strain SCAU265, capable of producing exopolysaccharides, was isolated from the gorgonian coral *Leptogorgia rigida* in the South China Sea (114°32′59 E, 22°40′37 N), according to a previous method by Liao et al. [17]. Morphological characteristics and genetic analysis based on internal transcribed spacer (ITS) rRNA sequences (accession number OR122480 in GenBank) were used to compare the strain with *A. pseudoglaucus* NRRL 40 (accession number NR135336.1) in GenBank (Fungi type and reference material), revealing a high similarity of 99.62%. Based on this analysis, the strain SCAU265 was identified as *A. pseudoglaucus* [17].

The fungal strain SCAU265 was cultivated in a 1000 mL Erlenmeyer flask containing 300 mL of medium. The medium consisted of the following components: 20 g/L maltose, 20 g/L mannitol, 20 g/L glucose, 10 g/L sodium glutamate, 3 g/L yeast extract, 1 g/L corn pulp, 0.5 g/L potassium dihydrogen phosphate, 0.3 g/L magnesium sulfate heptahydrate, and 30 g/L sea salt, with a pH of 7.2. The flask was placed on a rotary shaker and incubated at 26 °C for a duration of 7 days, with shaking at a speed of 150 rpm [18].

### 2.2. Extraction and Purification of ASP-1

After a 7-day incubation period, the entire fermentation broth (25.2 L, a total of 84 Erlenmeyer flasks were used for the fermentation) was filtered through cheesecloth and a Buchner funnel (with filter paper, pore size 30–50 μm) to remove the fungal mycelia. The filtered broth was then concentrated to 5 L under reduced pressure at a temperature of 55 °C. To precipitate the concentrated broth, ethanol (approximately four volumes of the concentrated broth) was added and the mixture was kept at 4 °C for 20 h. Subsequently, the mixture was centrifuged at 10,000 rpm for 10 min, resulting in the isolation of crude exopolysaccharides [19]. The crude exopolysaccharides were further purified by eliminating proteins using the Sevage method [20]. Subsequently, a dialysis bag (a molecular weight cutoff of approximately 4000 Da) was employed to remove small molecules such as sea salts. The crude polysaccharide ASP was fractionated on a Diethylaminoethyl-Sepharose-52 (DEAE-52) Fast Flow column (2.6 cm × 70 cm) by eluting with a step gradient of 0–1.2 mol/L NaCl solution at a flow rate of 1.0 mL/min. Eluate was collected by an auto-fraction collector (10 mL/tube). Each tube was tested for carbohydrate content by the classic phenol-sulfuric acid method with the detection wavelength of 490 nm [21]. The carbohydrates were first hydrolyzed into monosaccharides under the action of sulfuric acid, and rapidly dehydrated to form aldehyde derivatives. Then, they reacted with phenol to form orange–yellow compounds, which had a maximum absorption peak at a wavelength of 490 nm [21]. According to the profile of the gradient elution, the crude polysaccharide could be isolated two fractions. The fractions eluted with ultrapure water were pooled, dialyzed, and loaded onto a Sephadex G-75 column (2.6 cm × 60 cm), and eluted with ultrapure water at a flow rate of 0.4 mL/min. And then the major fractions were collected. Finally, the purified polysaccharide was freeze-dried and designated as ASP-1.

### 2.3. Purity and Molecular Weight of ASP-1

The purity and molecular weight of ASP-1 were determined using an HPGPC equipped with a BRT105-104-102 tandem column (0.8 cm × 30 cm) and a Shimadzu RI-10A detector, following a previously described method [22]. To determine the molecular weight of ASP-1, a standard dextran series and glucose were used as references.

### 2.4. General Analysis of ASP-1

The total sugar content of ASP-1 was determined using the phenol-sulfuric acid method, following the protocol described by DuBois et al. (1956) [21]. The sulfate content was measured using the barium chloride–gelatin method [23]. The protein content was quantified using the bicinchoninic acid (BCA) method [23]. Additionally, to assess the presence of nucleic acids and proteins, the ASP-1 solution (0.6 g/L) was analyzed using an ultraviolet–visible (UV-VIS) spectrophotometer (Cary 60, Agilent, Santa Clara, CA, USA) within the wavelength range of 190–400 nm.

To obtain the ASP-1 spectrum, a mixture of ASP-1 (1.28 mg) and dry KBr (100 mg) was prepared. The mixture was pressed into a pellet and scanned using a FTIR spectrum (Vector 33, Bruker, Billerica, MA, USA) in the range of 4000–400 cm^−1^ [18]. After being hydrolyzed with 3 mol/L Trifluoroacetic acid (TFA) solution at 120 °C for 3 h, the monosaccharide compositions of ASP-1 were determined using ion chromatography (ICS5000, ThermoFisher, New York, NY, USA) following the method described in a previous study [19].

### 2.5. Analysis of the Glycosidic Bonds between Residues of ASP-1

Analysis of the glycosidic bonds between residues of ASP-1 was conducted following Hakomori’s method [24]. In brief, ASP-1 was subjected to methylation, hydrolysis, and acetylation. The resulting product was then analyzed using a GC-MS Spectrometer equipped with MS column (30 mm × 0.25 mm × 0.25 μm) using programmed ramp-up conditions. The obtained results were compared with the Complex Carbohydrate Research Center Database (http://www.ccrc.uga.edu/, accessed on 3 September 2023) for analysis.

### 2.6. NMR Spectrum Analysis of ASP-1

To prepare for NMR analysis, ASP-1 was dissolved in D_2_O at a concentration of 60 mg/mL. The solution was then freeze-dried to facilitate deuterium exchange. Subsequently, the dried sample was fully dissolved in D_2_O and subjected to centrifugation [15]. The resulting supernatant was used for acquiring 1D NMR spectra (including ^1^H and ^13^C) as well as 2D NMR spectra (such as ^1^H-^1^H correlation spectroscopy (COSY), heteronuclear single quantum correlation (HSQC), and heteronuclear multiple bond correlation (HMBC)) using an NMR spectrometer (Bruker, Zurich, Switzerland).

### 2.7. Immunomodulatory Activity of ASP-1

The macrophage RAW 264.7 cell line (Procell, Wuhan, China)was utilized to assess the immunomodulatory activity of ASP-1. The RAW 264.7 cells were cultured in a humid incubator at 37 °C with 5% CO_2_, using Dulbecco modified eagle (DMEM) medium supplemented with 10% (*v*/*v*) fetal bovine serum (FBS) and 1% (*v*/*v*) penicillin–streptomycin [25].

The viability of RAW 264.7 cells was analyzed using a cell counting kit-8 (CCK-8) assay kit. RAW264.7 cells were seeded in 96-well plates at a density of 5 × 10^4^ cells/well in a humid incubator at 37 °C with 5% CO_2_ for 24 h. ASP-1 was then added to the wells at final concentrations of 0, 10, 50, 100, 200, 500, and 1000 μg/mL. After incubating for an additional 24 h, 10 μL of CCK-8 solution was added to each well and incubated for 2 h. The absorbance of each well was measured at 450 nm using a microplate reader [25].

RAW264.7 cells were cultured in 24-well plates at a density of 5 × 10^4^ cells/well in a humid incubator at 37 °C with 5% CO_2_ for 24 h. Subsequently, the cells were treated with ASP-1 at different concentrations (100, 200, and 400 μg/mL) and lipopolysaccharide (LPS) at a concentration of 2.5 μg/mL, respectively, and then further incubated for 24 h. The supernatant from each well was collected, and the levels of NO release, IL-6, and TNF-α secretion were determined using enzyme-linked immunosorbent assay (ELISA) kits [25].

### 2.8. Metabolomics Analysis

Metabolomics analysis was conducted to compare the effects of ASP-1 on RAW264.7 cells with a control group, following a previously established protocol [26]. RAW264.7 cells were treated with 500 μg/mL ASP-1, and a control group without treatment was included. The cells were collected for analysis, with each sample containing 1 × 10^7^ cells. An ultrahigh performance liquid chromatography (UPLC) system equipped with a 2.1 mm × 150 mm × 1.8 μm C_18_ column was used for sample separation and analysis. Mass spectrometry (MS) coupled with an electrospray ionization source (ESI) was employed to collect data in both positive and negative ion modes. Subsequently, pareto-scaled principal component analysis (PCA) and orthogonal partial least-squares discriminant analysis (OPLS-DA) were performed to analyze the obtained data. Variable importance in projection (VIP) values were calculated based on the OPLS-DA results. Metabolites showing a *p*-value < 0.05 and VIP > 1 were considered significant differently accumulated metabolites (DAMs). Further analysis focused on the DAMs, including identification of Kyoto Encyclopedia of Genes and Genomes (KEGG) pathways with a *p*-value < 0.05. Analyses were generated using R (Version 4.0.2), MetaboAnalyst (Version 4.0) [27] and KEGG Mapper (Version 5, released in July 2021) software [28].

### 2.9. Statistical Analysis

The data are expressed as the mean ± standard deviation (SD, *n* = 3). Statistical analysis was conducted using SPSS 26.0 software, employing one-way analysis of variance (ANOVA) or Student’s *t*-test. A significance level of *p*-value < 0.05 or *p*-value < 0.01 was considered statistically significant [25].

## 3. Results

### 3.1. Preparation and Physicochemical of ASP-1

The yield of ASP was 26.77 g, corresponding to a concentration of 1.07 g/L. ASP was isolated and purified using DEAE-52 (Figure 1a) and G-75 (Figure 1b) chromatographic columns, and the purified polysaccharides ASP-1 (the first peak in Figure 1b, 83.13% by weight) and ASP-2 (the second peak in Figure 1b, 16.87% by weight) were obtained. ASP-2 was not considered in the next experiment because the amount of ASP-2 was too small to measure the bioactivity. ASP-1 exhibited a total sugar content of 89.31 ± 0.70% and showed no presence of protein and nucleic acid, as confirmed by UV absorption at 260 nm and 280 nm (Figure 1c). Additionally, the Barium chloride-gelatin process indicated the absence of SO^3−^. The molecular weight of ASP-1 was determined to be 36.07 kDa based on the HPGPC analysis (Figure 1d).

### 3.2. Structural Characterization of ASP-1

The monosaccharide analysis of ASP-1 demonstrated that it primarily consisted of glucose (95.3%), with smaller amounts of mannose (3.9%) and galactose (0.8%). This composition suggests that ASP-1 likely possesses a main component of glucose (as illustrated in Figure 1e).

The FTIR analysis (Figure 1f) of ASP-1 exhibited characteristic absorption bands associated with polysaccharides. These included a broadband at 3369 cm^−1^ corresponding to the stretching vibration of O-H, and a band at 2904 cm^−1^ representing the stretching vibration of C-H. The weak absorption peak observed at 1642 cm^−1^ could be attributed to the binding of water molecules within the ASP-1 polymer [29]. The absorption bands at 1153 cm^−1^, 1079 cm^−1^, and 1053 cm^−1^ were indicative of C-O-C and C-O-H stretching vibrations, suggesting the presence of a pyran ring structure in ASP-1 [30]. The bands at 931 cm^−1^ and 848 cm^−1^ suggested the presence of α-glycosidic linkages in ASP-1. Additionally, the absence of an absorption peak at 1730 cm^−1^ indicated the absence of uronic acid in ASP-1 [29].

Methylation analysis (Table 1) of ASP-1 revealed the presence of ten linkages, including T(terminal)-Glc*p*, T-Man*p*, T-Gal*p*, 1,2-Man*p*, 1,3-Glc*p*, 1,4-Glc*p*, 1,6-Glc*p*, 1,3,4-Glc*p*, 1,4,6-Glc*p*, and 1,4,6-Gal*p*. These findings suggest that ASP-1 may have a main structure consisting of (1→4)-linked α-D-Glc*p* residues, with highly branched regions containing 1,4,6-glucopyranosyl (Glc*p*), 1,3,4-Glc*p*, 1,4,6-Gal*p*, T-Glc*p*, T-mannopyranosyl (Man*p*), and T-galactopyranosyl (Gal*p*) residues, thus forming a heteropolysaccharide structure.

To determine the linkages of the ten monosaccharide residues (A–J) identified through methylation analysis, NMR analysis was conducted (Figure 2a,b). The ^1^H NMR chemical shifts of the proton signals at δ_H_ values of 4.89 (H-1 in A), 3.69 (H-6 in A), 3.58 (H-5 in A), 3.58 (H-3 in A), 3.52 (H-2 in A), and 3.34 (H-4 in A), along with the ^13^C NMR chemical shifts at δ_C_ values of 98.57 (C-1 in A), 73.03 (C-5 in A), 72.86 (C-3 in A), 72.75 (C-4 in A), 71.66 (C-2 in A), and 60.52 (C-6 in A) were analyzed. The ^1^H-^1^H COSY and HSQC spectra (Figure 2c,d) enabled the differentiation of spin systems, including H-1/H-2, H-2/H-3, H-3/H-4, H-4/H-5, and H-5/H-6 in ASP-1. Based on these data, residue A was confirmed to be a T-α-Glcp residue [31]. Similarly, the other monosaccharide residues (B-J) were identified as follows: T-β-Manp (B) [32], T-α-Galp (C) [33], 1,2-α-Manp (D) [19], 1,3-α-Glcp (E) [34], 1,4-α-Glcp (F) [35], 1,6-β-Glcp (G) [36], 1,3,4-α-Glcp (H) [37], 1,4,6-α-Glcp (I) [35], and 1,4,6-α-Galp (J) [38].

The residues A-J were further structurally assigned and sequenced based on key HMBC correlations (Figure 2e). These correlations included H-1 (δ_H_ at 4.89, A) to C-4 (δ_C_ at 70.35, H), H-1 (δ_H_ at 4.91, H) to C-6 (δ_C_ at 70.35, G), H-1 (δ_H_ at 4.81, G) to C-4 (δ_C_ at 73.86, J), H-1 (δ_H_ at 5.27, D) to C-4 (δ_C_ at 73.86, J), H-1 (δ_H_ at 4.89, I) to C-4 (δ_C_ at 73.28, I), H-1 (δ_H_ at 5.33, F) to C-6 (δ_C_ at 70.30, I), H-1 (δ_H_ at 5.27, E) to C-4 (δ_C_ at 76.72, F), H-1 (δ_H_ at 5.28, J) to C-2 (δ_C_ at 77.47, D), H-1 (δ_H_ at 5.33, F) to C-4 (δ_C_ at 76.72, F), H-3 (δ_H_ at 3.87, H) to C-1 (δ_C_ at 98.57, I), H-6 (δ_H_ at 3.75, I) to C-1 (δ_C_ at 98.57, A), H-6 (δ_H_ at 3.75, I) to C-1 (δ_C_ at 98.57, I), H-4 (δ_H_ at 3.75, I) to C-1 (δ_C_ at 98.57, I), and H-4 (δ_H_ at 3.75, I) to C-1 (δ_C_ at 98.60, B). Utilizing COSY, HSQC, and HMBC data, the possible structure of ASP-1 is illustrated in Figure 2f, and the detailed ^1^H and ^13^C NMR chemical shifts of ASP-1 are assigned and presented in Table 2.The original mages (data) of ^1^H NMR, ^13^C NMR, ^1^H-^1^H COSY, HSQC, and HMBC are provided in the Appendix A.

### 3.3. Immunomodulatory Activity of ASP-1

RAW 264.7 macrophages were employed as an in vitro model to examine the potential effects of ASP-1 on cell proliferation and morphology. ASP-1 demonstrated the ability to enhance the proliferation of RAW 264.7 cells when treated with concentrations ranging from 10 to 1000 μg/mL. The promotion of cell proliferation by ASP-1 exhibited a concentration-dependent relationship, with a greater effect observed at higher ASP-1 concentrations. Notably, within the concentration range of 10 to 1000 μg/mL, the overall cell viability remained at or above 100%, indicating that ASP-1 did not exhibit cytotoxicity towards RAW 264.7 macrophages (Figure 3a). In the control group, the majority of cells exhibited a plump or round morphology with a smooth cell surface, while only a small proportion of spindle-shaped cells were undergoing proliferation and division (Figure 3b). However, in the experimental group treated with 400 μg/mL of ASP-1, there was an increased presence and proportion of spindle-shaped cells in the proliferation and division stage (Figure 3c).

During the immune response process, the production of active substances plays a vital role in immune regulation. Nitric oxide (NO) is one such active substance involved in immune regulation, and its release serves as an indicator of immune modulation. When assessing the immune regulatory potential of a substance, measuring NO release of RAW 264.7 macrophages can provide valuable insights. In this study, treatment with low concentrations of ASP-1 (100 and 200 μg/mL) did not significantly affect NO release compared to the blank control group. At a dose of 400 μg/mL, ASP-1 demonstrated a significant promotion of NO release (*p*-value < 0.001), but the NO release level was lower than that in the positive control group (2.5 μg/mL LPS) (Figure 4a). The noteworthy increase in NO release suggested that ASP-1 possessed potent immune-stimulating effects on RAW 264.7 macrophages.

The release of cytokines TNF-α and IL-6 was assessed using ELISA kits. When RAW 264.7 macrophages were treated with ASP-1 at concentrations of 100, 200, and 400 μg/mL for 24 h, the secretion of TNF-α increased significantly, presenting a clear dose-dependent pattern (Figure 4b). Furthermore, at a concentration of 400 μg/mL, the secretion of TNF-α surpassed that of the positive control group treated with LPS (Figure 4b) (*p*-value < 0.001). As for the effect of ASP-1 on IL-6 secretion, it was found that ASP-1 with high concentration (400 μg/mL) could significantly increase IL-6 secretion (*p*-value < 0.001), while the low concentrations (100 and 200 μg/mL) of ASP-1 had no significant effect on IL-6 secretion (Figure 4c).

### 3.4. Potential Immunomodulatory Mechanism of ASP-1

The metabolomics of macrophage RAW 264.7 cells treated with ASP-1 and control groups were performed using UPLC and MS-ESI systems. A total of 4054 metabolites were obtained in the treatment and control groups. The PCA highlighted the characteristic of differences in treatment and control groups. PC1 and PC2 explained 43.8% and 16.2% of variance in PCA score plot, respectively (Appendix A), indicating significant differences of the metabolites among treatment and control groups. The result was also supported by OPLS-DA (Appendix A). A total of 64 metabolites were identified as significantly different, with 28 upregulated and 36 downregulated metabolites. Among these significantly different metabolites, amino acids, peptides, and analogues (such as citrulline, glutamic acid, aspartic acid, and isoleucine, etc.) exhibited notable changes following ASP-1 treatment (Appendix A). The most prominently affected amino acid was citrulline, which increased by 86.22-fold. Furthermore, fatty acids and conjugates, and carbohydrates and carbohydrate conjugates also showed significant changes (Appendix A). For examples, itaconic acid, undecanoic acid, D-galactose, and trans-1,2-cyclohexanediol were increased by 21.58-, 4.09-, 3.82-, and 3.37-fold, respectively; while mannitol and sorbitol were decreased to 0.23- and 0.27-fold, respectively (Appendix A). In addition, other types of metabolites were relatively less distributed and had not been further analyzed in this study.

A KEGG enrichment analysis was conducted on the 64 differential metabolites that were identified, revealing 67 enriched signal pathways (*p*-value < 0.05). These pathways included 10 related to amino acid biosynthesis and metabolism, nine involved in signal transduction and membrane transport, eight involved in carbohydrate metabolism, five involved in energy metabolism, four associated with cofactors and vitamins, two related to nucleotide metabolism, and two involved in lipid metabolism. Among the top 20 significantly different metabolic pathways (Figure 5), the majority exhibited significant differences in amino acid metabolism (Appendix A). Further analysis of the significant differential metabolites within each pathway highlighted the frequent enrichment of metabolites such as citrulline, glutamic acid, aspartic acid, and isoleucine. This indicates that changes in amino acid metabolism are the most prominent metabolic alterations in RAW264.7 macrophages treated with ASP-1.

## 4. Discussion

Marine fungal polysaccharides possess many advantages in terms of their broad sources, distinctive structures, and significant health effects [39,40], making them highly promising for immune-related activities [5,6]. Numerous experiments have confirmed their immunomodulatory functions on macrophages [41]. However, most studies have focused on investigating the immune activity of marine fungal polysaccharides on macrophages and identifying the factors that influence their effects. Consequently, there is a relative scarcity of research on the molecular mechanisms underlying how marine fungal polysaccharides regulate the release of immune factors from macrophages. In this study, we present for the first time the structural characterization, immunomodulatory activity, and potential mechanism of a novel exopolysaccharide derived from *A*. *pseudoglaucus* SCAU265, isolated from a gorgonian coral *L. rigida* in the South China Sea.

Several recent studies have focused on the unique structure of marine fungal polysaccharides [42,43]. However, the structure of ASP-1 differed from the polysaccharides produced by other marine-derived fungal isolates previously reported. For example, the polysaccharide from coral-derived *A. versicolor*, as reported by Chen et al. [4], contained a →6)-α-D-Glc*p*(1→residue. On the other hand, the polysaccharide from deep-sea-derived *A. versicolor*, as reported by Yan et al. [22], contained→2,6)-α-D-Glc*p*(1→ and →2)-β-D-Glc*p*-(1→residues. In contrast, ASP-1 had a main structure consisting of (1→4)-linked α-D-Glc*p* residues, along with 1,4,6-Glc*p*, 1,3,4-Glc*p*, 1,4,6-Gal*p*, T-Glc*p*, T-Man*p*, and T-Gal*p* residues incorporated in the highly branched heteropolysaccharide. Interestingly, four 1,4,6-α-Glc*p* residues of ASP-1 were connected together to form a square structure (Figure 2f), which was relatively rare in marine fungal extracellular polysaccharides. The diverse chemical structures exhibited by these marine fungal polysaccharides can be attributed to the wide range of marine environments that foster diverse marine fungi.

Immunological activity is a common and well-documented biological characteristic of marine fungal polysaccharides. In this study, it was observed that the polysaccharide ASP-1 exhibited a strong immune-stimulating effect on macrophage RAW 264.7 cells, as evidenced by a significant increase in NO release. This finding aligns with the immune activity demonstrated by another marine fungal polysaccharide, AVP141-A, which was previously reported [13]. Moreover, at a concentration of 400 μg/mL, ASP-1 could significantly enhance the production of TNF-α and IL-6 in RAW264.7 macrophages, which was consistent with a previous relevant report [44]. Tian et al. obtained an exopolysaccharide EPS1 from fungal isolate *Paecilomyces cicadae* TJJ1213 and found that EPS1 (with the concentration of 400 μg/mL) could significantly up-regulate the mRNA expression of TNF-α and IL-6 in macrophage RAW 264.7 cells [44]. These results collectively highlight the immense potential of marine fungal polysaccharides in terms of immune activity.

Although the corresponding relationship was not clearly elucidated, the monosaccharide composition, structure, and molecular weight of ASP-1 might be related to its immunological activity. Compared to β-glucans, less attention had been paid to α-glucans on the immunological activity, but recently many exopolysaccharides with α-glucans had been reported to exhibit immunological activity, such as ASP-1 (this study) and CEPSN-1from *Auricularia auricula-judae* [45]. Similar to exopolysaccharide CEPSN-1, polysaccharide ASP-1 was also mainly composed of glucose, mannose, and galactose, but its immune activity (action concentration and molecular weight were 400 μg/mL and 36.07 kDa, respectively)was lower than CEPSN-1 (action concentration and molecular weight were 200 μg/mL and 4.60 kDa, respectively), which was similar to previous reports supporting that there was a negative correlation between molecular weight and biological activity [45,46].

Metabolomics analysis provided insights into the potential immunomodulatory molecular mechanism of ASP-1, suggesting that it primarily influences amino acid anabolism to regulate macrophage immune function. Amino acids serve as fundamental building blocks of proteins and fulfill various functions, including signal transmission, nitrogen balance, participation in nucleotide synthesis, and involvement in the biosynthesis of antioxidants (such as glutathione), glucosamine, and polyamines [47]. Remarkable changes in amino acid levels were observed in the metabolite profile following ASP-1 treatment, with the most notable increase observed in citrulline, showing an 86.22-fold change (Appendix A). Citrulline plays a significant role in the physiological and biochemical reactions within the body. It serves as a precursor for several amino acids, including proline, glutamine, and guanidine, and contributes to improving the physiological and metabolic functions of organisms, including enhancing immune function [48]. As the precursor of glutamine, glutamic acid acts as an important immune regulator [49]. It can stimulate the mitosis, differentiation, and proliferation of macrophages, as well as promote cytokine secretion [50]. The increased levels of these amino acids likely correspond to an enhancement in the immune function of immune cells.

Among the downregulated amino acids, aspartic acid and isoleucine did not exhibit an obvious decrease (Appendix A), but they displayed a significant difference in amino acid metabolism (Appendix A). Aspartic acid has a role in the urea cycle process, where it can react with citrulline to produce arginine [51]. Hence, the decrease in aspartic acid levels in this study might be related to the reduced production of arginine through the urea cycle process. Arginine, an essential amino acid, serves as a precursor for the synthesis of various substances such as NO, proline, and guanidine [52]. It is known that arginine, as a precursor of NO, plays a regulatory role in immune function through enzymatic processes [52]. Previous research by Sahana et al. demonstrated that a polysaccharide produced by the marine microorganism *Alteromonas* sp. PRIM-28 increased the level of NO in macrophages and upregulated arginase expression [53]. Moreover, exogenous supplementation of L-arginine has been shown to promote NO release [54]. In the present study, it was observed that treatment with ASP-1 at a concentration of 400 μg/mL significantly enhanced NO release (Figure 4a), which corresponded to a decrease in the metabolite arginine (Appendix A). Arginine can be metabolized to produce proline and glutamic acid, with proline playing a crucial role in immune regulation [55,56].

## 5. Conclusions

In conclusion, this study explored the properties of ASP-1, a novel water-soluble fungal exopolysaccharide derived from the marine coral-associated fungus *A*. *pseudoglaucus* SCAU265. ASP-1 exhibited a molecular weight of 36.07 kDa and was predominantly composed of glucose (95.3%), mannose (3.9%), and galactose (0.8%) residues. Its structural characterization revealed a mainly composed of (1→4)-linked α-D-Glc*p* residues, accompanied by highly branched heteropolysaccharide regions containing 1,4,6-Glc*p*, 1,3,4-Glc*p*, 1,4,6-Gal*p*, T-Glc*p*, T-Man*p*, and T-Gal*p* residues. In vitro immunomodulatory assays demonstrated that ASP-1 exhibited remarkable immune-enhancing activity, as indicated by the significant elevation of NO, TNF-α, and IL-6 levels in RAW264.7 macrophages. Metabolomics analysis shed light on the potential mechanism underlying ASP-1’s immunomodulatory effects, revealing its influence on amino acid anabolism, particularly in arginine synthesis and metabolism. These findings provide valuable scientific groundwork for further investigations into the polysaccharide derived from *A*. *pseudoglaucus* SCAU265. Future studies will focus on verifying the immunomodulatory activity and mechanism in vivo, with the ultimate goal of developing and utilizing ASP-1, the marine fungus-derived exopolysaccharide, as a promising immune activator.

## Figures and Tables

**Figure 1 jof-09-01057-f001:**
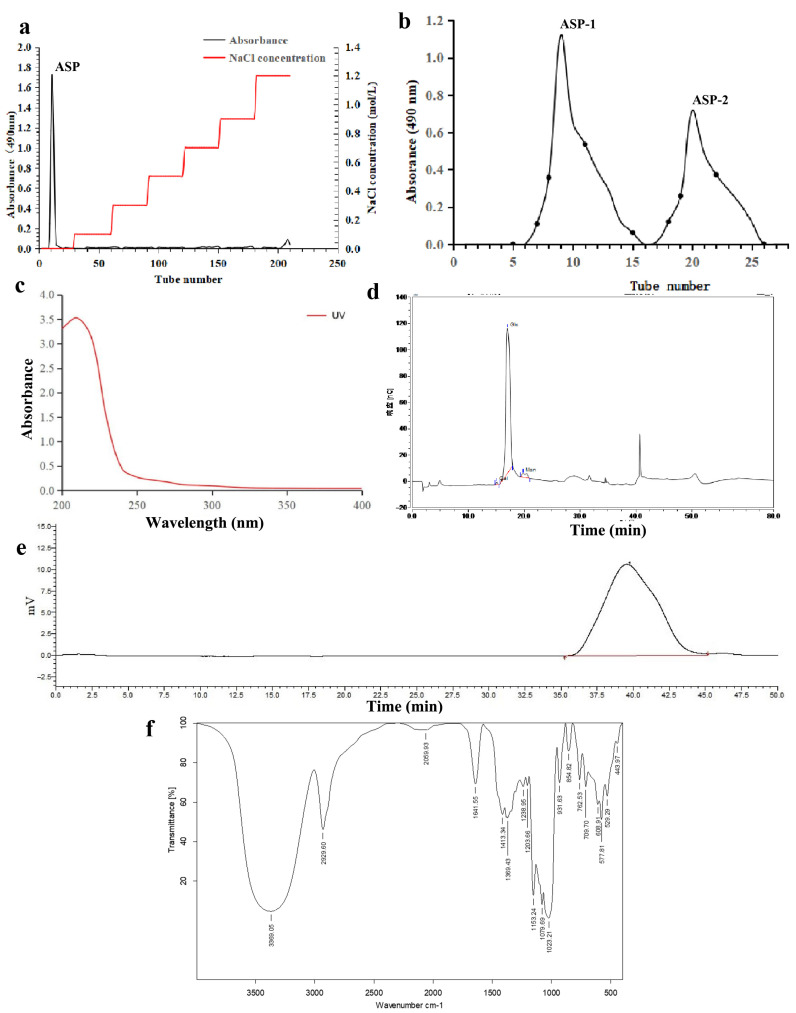
ASP-1 elution chromatogram on DEAE-52 Fast Flow (**a**) and Sephadex G-75 (**b**), UV spectrum of ASP-1 (**c**), monosaccharides of ASP-1 by Ion Chromatogram (**d**), HPGPC chromatogram of ASP-1 (**e**), and FT-IR spectrum of ASP-1 (**f**).

**Figure 2 jof-09-01057-f002:**
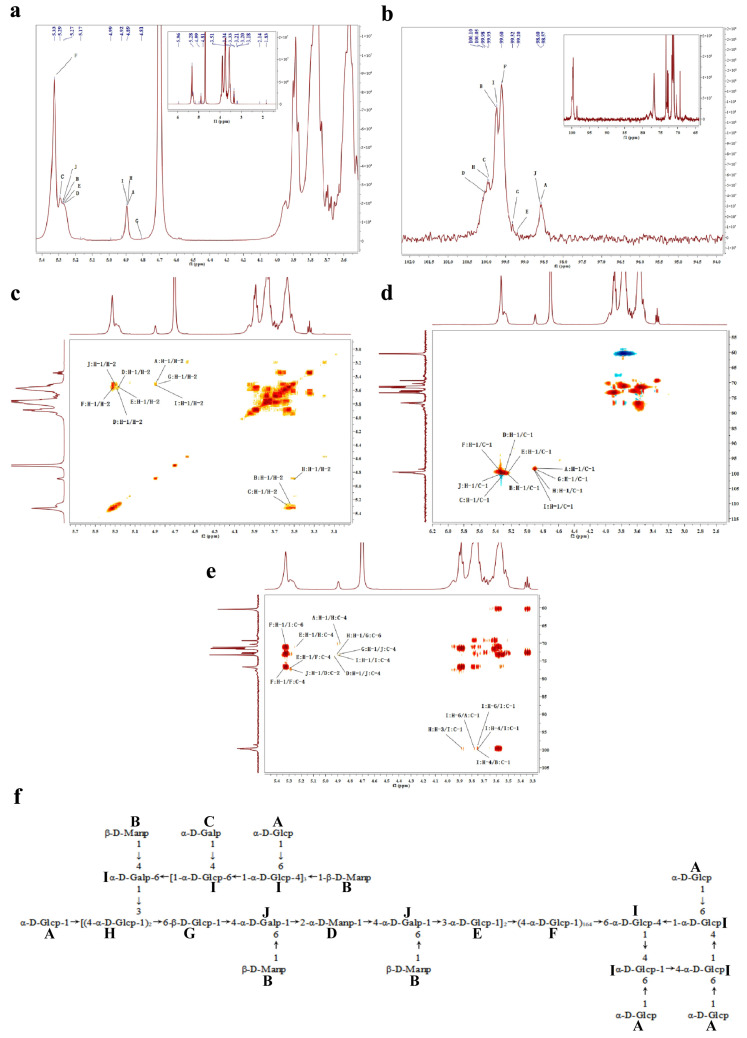
NMR analysis of ASP-1; ^1^H NMR(**a**); ^13^C NMR (**b**); ^1^H-^1^H COSY (**c**); HSQC (**d**); HMBC (**e**); the chemical structure of ASP-1 (**f**), Residue A: T(terminal)-α-Glc*p*, Residue B: T-β-Man*p*, Residue C: T-α-Gal*p*, Residue D: 1,2-α-Man*p*, Residue E: 1,3-α-Glc*p*, Residue F: 1,4-α-Glc*p*, Residue G: 1,6-β-Glc*p*, Residue H: 1,3,4-α-Glc*p*, Residue I: 1,4,6-α-Glc*p*, Residue J: 1,4,6-α-Gal*p*.

**Figure 3 jof-09-01057-f003:**
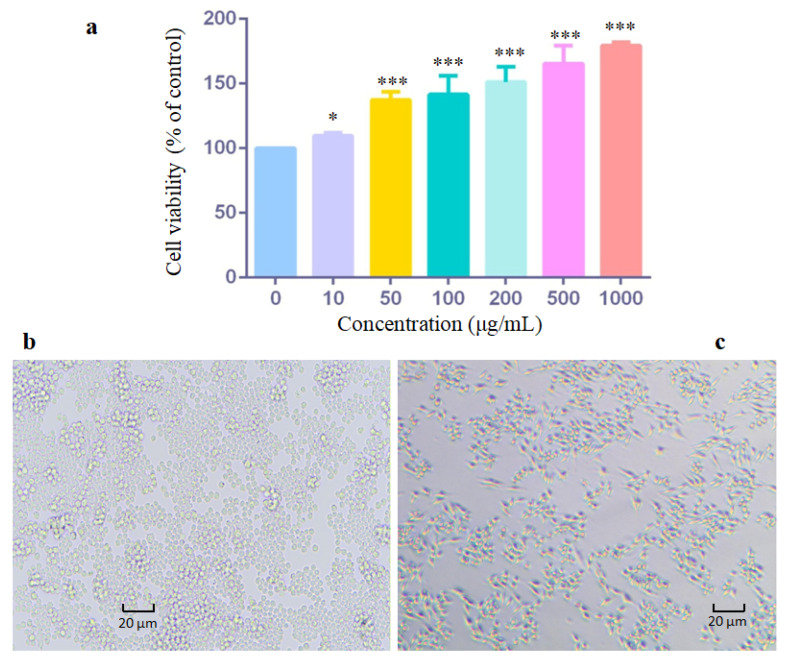
Cell viability (**a**) and morphological changes of RAW 264.7 cells treated with different concentrations of ASP-1 for 24 h ((**b**): control group, (**c**): 400 μg/mL). The data were presented as means ± SD. Compared to control group as *: 0.05 > *p*-value > 0.01, ***: *p*-value < 0.001 vs. control group.

**Figure 4 jof-09-01057-f004:**
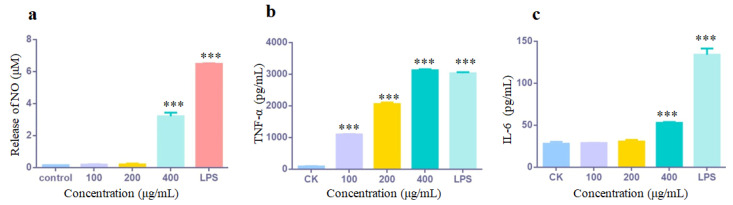
Effects of ASP-1 on the release of NO (**a**), TNF-α (**b**) and IL-6 (**c**). The data were presented as means ± SD. Compared to control group as *** *p*-value < 0.001.

**Figure 5 jof-09-01057-f005:**
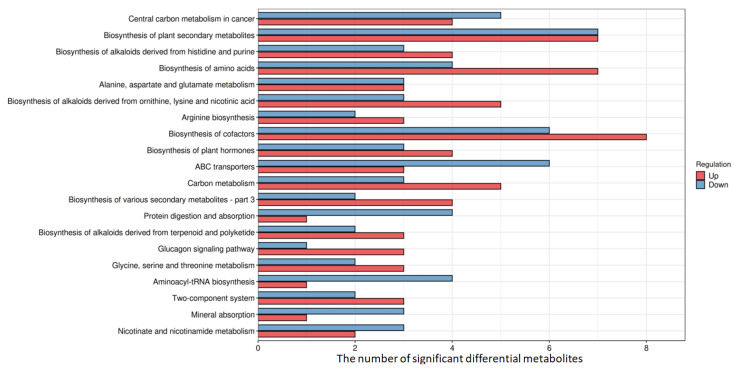
Histogram of top 20 enrichment of differential metabolic pathways.

**Table 1 jof-09-01057-t001:** Methylation analysis of ASP-1.

Peak	Retention Time (min)	Methylated Sugars	Deduced Residues	Molar Ratio (%)	Major Mass Fragments (*m*/*z*)
A	17.036	2,3,4,6-Me4-Glc*p*	T-Glc*p*	0.064	43,71,87,101,117,129,145, 161,205
B	17.641	2,3,4,6-Me4-Man*p*	T-Man*p*	0.031	43,71,87,101,117,129,145, 161,205
C	18.108	2,3,4,6-Me4-Gal*p*	T-Gal*p*	0.045	43,71,87,101,117,129,145, 161,205
D	20.872	3,4,6-Me3-Man*p*	1,2-Man*p*	0.008	43,87,129,161,189
E	21.13	2,4,6-Me3-Glc*p*	1,3-Glc*p*	0.014	43,87,99,101,117,129,161, 173,233
F	21.485	2,3,6-Me3-Glc*p*	1,4-Glc*p*	0.657	43,87,99,101,113,117,129, 131,161,173,233
G	23.773	2,3,4-Me3-Glc*p*	1,6-Glc*p*	0.013	43,87,99,101,117,129,161, 189,233
H	24.528	2,6-Me2-Glc*p*	1,3,4-Glc*p*	0.021	43,87,97,117,159,185
I	26.763	2,3-Me2-Glc*p*	1,4,6-Glc*p*	0.128	43,71,85,87,99,101,117,127, 159,161,201
J	28.63	2,3-Me2-Gal*p*	1,4,6-Gal*p*	0.018	43,71,85,87,99,101,117,127, 159,161,201,261

**Table 2 jof-09-01057-t002:** Chemical shifts of ASP-1 residues.

Glycosyl Residues	ppm (H/C)	1	2	3	4	5	6
A: T-α-Glc*p*	H	4.89	3.52	3.58	3.34	3.58	3.69
	C	98.57	71.66	72.86	72.75	73.03	60.52
B: T-β-Man*p*	H	5.27	3.55	3.64	3.87	4.08	3.46
	C	99.98	77.92	72.63	69.98	70.55	61.87
C: T-α-Gal*p*	H	5.29	3.58	3.83	4.19	3.93	3.72
	C	99.93	68.00	69.33	70.75	68.75	60.42
D: 1,2-α-Man*p*	H	5.27	3.56	3.92	3.79	3.64	3.75
	C	100.01	77.47	68.95	65.20	71.77	60.04
E: 1,3-α-Glc*p*	H	5.27	3.55	3.89	3.87	3.57	3.84/3.66
	C	99.20	73.31	78.74	71.5	71.15	60.43
F: 1,4-α-Glc*p*	H	5.33	3.55	3.73	3.58	3.93	3.69
	C	99.60	71.96	72.22	76.72	71.08	60.86
G: 1,6-β-Glc*p*	H	4.81	3.50	4.61	3.51	3.94	3.60
	C	99.32	70.70	73.40	70.40	71.33	73.07
H: 1,3,4-α-Glc*p*	H	4.91	3.51	3.87	3.51	3.83	3.88
	C	99.98	71.41	78.11	70.35	70.48	65.69
I: 1,4,6-α-Glc*p*	H	4.89	3.57	3.65	3.75	3.85	3.75
	C	98.57	75.73	74.02	73.28	74.60	70.30
J: 1,4,6-α-Gal*p*	H	5.28	3.54	3.81	3.57	3.74	3.60
	C	98.60	74.53	76.24	73.86	74.50	60.38

## Data Availability

Not applicable.

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
