# Peer review of "A New Exopolysaccharide of Marine Coral-Associated Aspergillus pseudoglaucus SCAU265: Structural Characterization and Immunomodulatory Activity"

_jof, 2023, doi:10.3390/jof9111057_

Round 1

Reviewer 1 Report

Thank you for the opportunity to read your good science. 

A new exopolysaccharide of marine coral-associated Aspergillus sp. SCAU265: Structural characterization and immunomodulatory activity

Bo Peng * , Yongchun Liu , Yuqi Lin , Supaluck Kraithong , Li Mo , Ziqing Gao , Riming Huang , Xiaoyong Zhang *

Overall, this is a good study worth publishing. Please address the following changes.

Pg      Lines

1        16 – 18                   There are grammar issues with this sentence. It is awkward.

1        35 - 36                    The statement of “the investigation of their symbiotic microorganisms, which may produce these reported metabolites, remains limited” is simply untrue.  This is the major focus of many studies in the last 20 years since it was discovered that symbionts are the likely source of structurally complex secondary metabolites.

2        70 – 80                   This information belongs in the methods.  What is missing here is the OBJECTIVES of the study or the HYPOTHESIS that is being tested.

2        83                Was it already known that this strain was “capable of producing exopolysaccharides”?  How did the authors obtain this information prior to making their collections.

2        98                What are the authors referring to as “impurities”?  Is it cellular material?  Cheesecloth cannot remove chemical impurities.

4        178 – 189     The methods described here are redundant with the section above.  This paragraph can be streamlined and should be.

Results section:  the majority of the NMR data should be included in the supplemental materials unless

9        268 – 269 and 279 – 280   These statements are too similar and appear redundant to the reader.

11       323               What do the authors mean by unique advantages?  Perhaps there is a better term to be used here.

12      371               Typo – information

Overall it is good. There were some cases where the wrong words were used. 

Author Response

Dear reviewer (1),

Journal of Fungi,

On behalf of my co-authors, I would like to submit the revised manuscript entitled “A new exopolysaccharide of marine coral-associated Aspergillus sp. SCAU265: Structural characterization and immunomodulatory activity” (jof-2615601 to Journal of Fungi.

The authors are grateful to the reviewers for their valuable comments and time, and all comments are responded carefully point by point.

**********************************

Overall, this is a good study worth publishing. Please address the following changes.

Response: The author greatly appreciates your positive comments, and all comments are responded carefully point by point.

  1. Page 1, Lines 16-18:  There are grammar issues with this sentence. It is awkward.

Response: The sentence had been revised, which was as follows.

A novel water-soluble fungal exopolysaccharide ASP-1 was isolated from the fermentation broths of marine coral-associated fungus Aspergillus sp. SCAU265, and purified by Diethylaminoethyl-Sepharose-52 (DEAE-52) Fast Flow and Sephadex G-75.

  1. Page 1, Lines 35 - 36:  The statement of “the investigation of their symbiotic microorganisms, which may produce these reported metabolites, remains limited” is simply untrue.  This is the major focus of many studies in the last 20 years since it was discovered that symbionts are the likely source of structurally complex secondary metabolites.

Response: Deleted the statement of “the investigation of their symbiotic microorganisms, which may produce these reported metabolites, remains limited”. And the sentence had been revised as follows.

However, recent studies have found that many of these biological compounds are produced by their symbiotic microorganisms [2]. The study of biological compounds from symbiotic microorganisms in marine invertebrates is currently a hot topic of research.

The reference of [2] was also changed as follows.

[2] Paul, V.J.; Puglisi, M.P. Chemical mediation of interactions among marine organisms. Natural Product Reports 2004, 21, 189–209.

  1. Page 2, Lines 70 - 80: This information belongs in the methods.  What is missing here is the OBJECTIVES of the study or the HYPOTHESIS that is being tested.

Response: the following sentences were added into the paragraph.

In the present study, our objective was further investigating the potential application of exopolysaccharides from marine-derived Aspergillus spp., elucidating their structural properties and biological activities, particularly their immunomodulatory function..

  1. Page 2, Lines 83: Was it already known that this strain was “capable of producing exopolysaccharides”?  How did the authors obtain this information prior to making their collections.

Response: Yes, before conducting this work, we had screened the strain (including other fungal strains from the gorgonian coral Leptogorgia rigida in the South China Sea) for its ability to produce extracellular polysaccharides. And this strain showed a good result. Because we feel that this screening work is not too closely related to the content of this section, it was not described in the manuscript.

  1. Page 2, Lines 98: What are the authors referring to as “impurities”?  Is it cellular material?  Cheesecloth cannot remove chemical impurities.

Response: The so-called impurities here actually refer to the fungal mycelia after fungal fermentation. Extracellular polysaccharides are present in the fermentation broth, so we need to remove the fungal mycelia. Cheesecloth cannot remove chemical impurities, but it can filter the fungal mycelia.

Therefore, the words of “any impurities” were changed to “the fungal mycelia” in the revised manuscript.

  1. Page 4, Lines 178 – 189: (1) The methods described here are redundant with the section above.  This paragraph can be streamlined and should be.

(2) Results section:  the majority of the NMR data should be included in the supplemental materials unless.

Response:  (1) This paragraph had been streamlined, which was as follows.

The yield of ASP was 26.77 g, corresponding to a concentration of 1.07 g/L. ASP was isolated and purified using DEAE-52 (Fig. 1a) and G-75 (Fig. 1b) chromatographic columns, and the purified polysaccharides ASP-1 (the first peak in Fig. 1b, 83.13% by weight) and ASP-2 (the second peak in Fig. 1b, 16.87% by weight) were obtained. ASP-2 was not considered in the next experiment because the amount of ASP-2 was too small to measure the bioactivity. ASP-1 exhibited a total sugar content of 89.31±0.70% and showed no presence of protein and nucleic acid, as confirmed by UV absorption at 260 nm and 280 nm (Fig. 1c). Additionally, the Barium chloride-gelatin process indicated the absence of SO3-. The molecular weight of ASP-1 was determined to be 36.07 kDa based on the HPGPC analysis (Fig. 1d).

(2) The original mages (data) of 1H NMR, 13C NMR, 1H-1H COSY, HSQC and HMBC were provided in supplemental materials (Fig. s1-Fig. s5). Please check the supplemental materials.

  1. Page 9, Lines 268 – 269 and 279 – 280: These statements are too similar and appear redundant to the reader.

Response: Removed the sentences in Lines 279 – 280. Please check the revised manuscript.

  1. Page 11, Lines 323: What do the authors mean by unique advantages?  Perhaps there is a better term to be used here.

Response: The unique advantages (of marine fungal polysaccharides) mean wide range of sources, unique structure, significant health effects and few side effects. However, the words of “unique advantages” were not appropriate here, which was changed to “many advantages”.

  1. Page 12, Lines 371: Typo – information

Response: Changed “infornmation” to “information”. In the revised manuscript, the words “supporting information” were changed to “supplemental materials”.

**********************************

All revised contents in text were highlighted in red in the revised manuscript.

Thank you very much for your consideration of our manuscript for potential publication. We look forward to hearing from you soon.

Best Regards.

Sincerely yours,

Dr. Xiaoyong Zhang

University Joint Laboratory of Guangdong Province, Hong Kong and Macao Region on Marine Bioresource Conservation and Exploitation, College of Marine Sciences, South China Agricultural University, Guangzhou 510642, China

E-mail: zhangxiaoyong@scau.edu.cn

Reviewer 2 Report

Review for

 Article  

A new exopolysaccharide of marine coral-associated Aspergillus  sp. SCAU265: Structural characterization and  immunomodulatory activity

It is fully true that the study of marine microbial exopolysaccharides is of great interest.

Most of the research is done on bacterial EPS, so research about fungal EPS is welcome.

Previous authors focused on antioxidant activities:

Protection of PC12 cells from hydrogen peroxide-induced injury by EPS2, an exopolysaccharide from a marine filamentous fungus Keissleriella sp. YS4108

Sun, C.Shan, C.Y.Gao, X.D.Tan, R.X.

Journal of Biotechnology, 2005, 115(2), pp. 137–144

Free radical scavenging and antioxidant activities of EPS2, an exopolysaccharide produced by a marine filamentous fungus Keissleriella sp. YS 4108

Sun, C.Wang, J.-W.Fang, L.Gao, X.-D.Tan, R.-X.

Life Sciences, 2004, 75(9), pp. 1063–1073

-------------------------------------

large scale production of microbial EPS

is known for bacteria

curdlan by Alcaligenes faecalis var. myxogenes

xanthan by Xanthomonas campestris

what about industrial production from fungi?

--------------------------------------

 There is a relative scarcity of research on immunomodulatory activity of marine fungal exopolysaccharides. The present study investigates a novel water-soluble fungal exopolysaccharide ASP-1that was isolated and purified from the fermentation broths of a marine coral-associated fungus Aspergillus sp.

----------------------

strain not determined at the species level, why?

The fungal strain SCAU265, capable of producing exopolysaccharides, was isolated from the 83 gorgonian coral Leptogorgia rigida in the South China Sea (114°3259 E, 22°4037 N), according to a 84 previous method by Liao et al. [17]. Morphological characteristics and genetic analysis based on 85 internal transcribed spacer (ITS) rRNA sequences (accession number OR122480 in GenBank) were 86 used to compare the strain with Aspergillus sp. 37a1 (accession number MN912600) in GenBank, 87 revealing a high similarity of 99.82%. Based on this analysis, the strain SCAU265 was identified as 88 Aspergillus sp. [17].

authors are now using multigene phylogeny for fungi

·         Published: 07 June 2019

A multigene phylogeny toward a new phylogenetic classification of Leotiomycetes

--------------------------

similar study by same research group, on another strain

Article

Structural characterization and immunomodulatory activity of an exopolysaccharide from marine-derived Aspergillus versicolor SCAU141

Wu, K.Li, Y.Lin, Y., ...Huang, R.Zhang, X.

International Journal of Biological Macromolecules, 2023, 227, pp. 329–339

------------------------

true novelty of SCAU265 compared to SCAU141??

-----------------

Despite the extensive research on 56 secondary metabolites, there has been limited investigation into biological extracellular 57 polysaccharides derived from marine Aspergillus species. Some notable examples include the 58 exopolysaccharide AVP-141 from marine fungus Aspergillus versicolor SCAU141 [13], the 59 antioxidant extracellular polysaccharide YSS from A. terreus [14], the exopolysaccharide AWP 60 isolated from a marine coral-associated A. versicolor [15], and an antioxidant exopolysaccharide 61 isolated from a marine-derived Aspergillus sp. Y1 [16].

part of the answer

This finding aligns with the immune activity demonstrated by another 347 marine fungal polysaccharide, AVP141-A, which was previously reported [13]. Moreover, ASP-1 348 significantly enhanced the production of TNF-α and IL-6 in RAW264.7 macrophages, indicating its 349 capacity to promote the activation of M1 immune cells and thereby mediate the inflammatory 350 response [41].

minor revision

Author Response

Dear reviewer (2),

Journal of Fungi,

On behalf of my co-authors, I would like to submit the revised manuscript entitled “A new exopolysaccharide of marine coral-associated Aspergillus sp. SCAU265: Structural characterization and immunomodulatory activity” (jof-2615601 to Journal of Fungi.

The authors are grateful to the reviewers for their valuable comments and time, and all comments are responded carefully point by point.

**********************************

  1. (1) It is fully true that the study of marine microbial exopolysaccharides is of great interest. Most of the research is done on bacterial EPS, so research about fungal EPS is welcome.

 (2) Previous authors focused on antioxidant activities:

 Protection of PC12 cells from hydrogen peroxide-induced injury by EPS2, an exopolysaccharide from a marine filamentous fungus Keissleriella sp. YS4108. Sun, C., Shan, C.Y., Gao, X.D., Tan, R.X. Journal of Biotechnology, 2005, 115(2), pp. 137–144.

Free radical scavenging and antioxidant activities of EPS2, an exopolysaccharide produced by a marine filamentous fungus Keissleriella sp. YS 4108. Sun, C., Wang, J.-W., Fang, L., Gao, X.-D., Tan, R.-X.,Life Sciences, 2004, 75(9), pp. 1063–1073.

Response: (1) The author greatly appreciates your positive comments, and all comments are responded carefully point by point.

(2) Thank you for your strong recommendation. I have carefully read these two articles and cited them in the article. Which were as follows.

Marine fungal polysaccharides possess many advantages in terms of their broad sources, distinctive structures, and significant health effects [40, 41], making them highly promising for immune-related activities [5, 6].

[40] Sun, C.; Wang, J.; Fang, L.; Gao, X.; Tan, R. Free radical scavenging and antioxidant activities of EPS2, an exopolysaccharide produced by a marine filamentous fungus Keissleriella sp. YS 4108. Life Sciences 2004, 75, 1063–1073.

 [41] Sun, C.; Shan, C.Y.; Gao, X.D.; Tan, R.X.  Protection of PC12 cells from hydrogen peroxide-induced injury by EPS2, an exopolysaccharide from a marine filamentous fungus Keissleriella sp. YS4108. Journal of Biotechnology2005, 115, 137–144.

  1. Large scale production of microbial EPS is known for bacteria curdlan by Alcaligenes faecalis var.myxogenes xanthan, by Xanthomonas campestris, what about industrial production from fungi?

Response: Compared to bacterial polysaccharides, there are relatively few industrial products of fungal polysaccharides, and currently the more familiar ones are Ganoderma lucidum polysaccharides (Yang et al., Journal of Functional Foods, 2022,92:105069; Liu et al., Process biochemistry, 2022,119, 96-105; Camargo et al., Journal of Ethnopharmacology, 2022, 286, 114891) and Cordyceps militaris polysaccharides (Lin et al.,   African Health Sciences, 2019, 2, 2156-2163; Liu et al., International Journal of Biological Macromolecules, 2016, 86, 594-598; Gao et al., Advanced Materials Research, 2011, 287-290, 2003-2007).  

  1. There is a relative scarcity of research on immunomodulatory activity of marine fungal exopolysaccharides. The present study investigates a novel water-soluble fungal exopolysaccharide ASP-1that was isolated and purified from the fermentation broths of a marine coral-associated fungus Aspergillus sp.

Response: Yes, there is a relative scarcity of research on immunomodulatory activity of marine fungal exopolysaccharides. A novel water-soluble fungal exopolysaccharide ASP-1 was isolated from the fermentation broths of marine coral-associated fungus Aspergillus sp. SCAU265, and purified by Diethylaminoethyl-Sepharose-52 (DEAE-52) Fast Flow and Sephadex G-75. Structural analysis revealed that ASP-1 had an average molecular weight of 36.07 kDa and was mainly composed of (1→4)-linked α-D-glucopyranosyl residues, along with highly branched heteropolysaccharide regions containing 1,4,6-glucopyranosyl, 1,3,4-glucopyranosyl, 1,4,6- galactopyranosyl, T-glucopyranosyl, T-mannopyranosyl, and T-galactopyranosyl residues. ASP-1 demonstrated significant effects on the proliferation, nitric oxide levels, and secretion of cytokines TNF-α and IL-6 in macrophage RAW264.7 cells. Metabolomics analysis provided insights into the potential mechanisms of immune regulatory of ASP-1, suggesting its involvement in regulating immune function by modulating amino acid anabolism, particularly arginine synthesis and metabolism. These findings provide fundamental scientific data for further research on its accurate molecular mechanism of immunomodulatory activity.

  1. (1) Strain not determined at the species level, why?  

The fungal strain SCAU265, capable of producing exopolysaccharides, was isolated from the  gorgonian coral Leptogorgia rigida in the South China Sea (114°32′59 E, 22°40′37 N), according to a  previous method by Liao et al. [17]. Morphological characteristics and genetic analysis based on  internal transcribed spacer (ITS) rRNA sequences (accession number OR122480 in GenBank) were  used to compare the strain with Aspergillus sp. 37a1 (accession number MN912600) in GenBank, revealing a high similarity of 99.82%. Based on this analysis, the strain SCAU265 was identified as  Aspergillus sp. [17].

(2) authors are now using multigene phylogeny for fungi,  Published: 07 June 2019, A multigene phylogeny toward a new phylogenetic classification of Leotiomycetes,· Peter R. Johnston, et al. IMA Fungus volume 10, Article number: 1 (2019).

Response: Yes, the strain SCAU265 was not determined at the species level based on internal transcribed spacer (ITS) rRNA sequences. When the ITS sequence (accession number OR122480 in GenBank) of the strain was compared on Genbank, the first three strains with the highest similarity were all genus level strains. Please check the result (Figure a1).

Figure a1. Alignment result of ITS sequence of the strain SCAU265

(2) After reading the literature (A multigene phylogeny toward a new phylogenetic classification of Leotiomycetes, Peter R. Johnston, et al. IMA Fungus volume 10, Article number: 1 (2019).) you provided, I found that the limitation of our article is that we only use ITS sequences for molecular biological identification of fungal strains.

In future research, we will try to consider using multiple genes for fungal strain identification. Thank you very much for providing this suggestion.

  1. Similar study by same research group, on another strain. Article, Structural characterization and immunomodulatory activity of an exopolysaccharide from marine-derived Aspergillus versicolor SCAU141, Wu, K., Li, Y., Lin, Y., ...Huang, R., Zhang, X., International Journal of Biological Macromolecules, 2023, 227, pp. 329–339.

Response: The article mentioned above is similar study by same research group, on another strain. However, it has significant differences or characteristics from current research.

Firstly, the sources of the strains are different, one (Aspergillus versicolor SCAU141) was from the Scleractinia coral, and the other (Aspergillus sp. SCAU265) was from gorgonian coral. Additionally, and most importantly, the structure of fungal polysaccharides varies greatly, with different monosaccharide compositions and molecular weights. The most unique feature was that the polysaccharide AVP-141-1 also contained sulfate ester (approximately 3.62 %), while the polysaccharide ASP-1 did not contain sulfate ester.

  1. True novelty of SCAU265 compared to SCAU141?

Response: Yes, compared to SCAU141, the strain SCAU265 is novelty and different.

The strain SCAU141 was isolated from Scleractinia coral. It was identified as Aspergillus versicolor by morphological and internal transcribed spacer (ITS) rRNA sequences, and the sequence data were deposited in NCBI under accession number MF135504.

Compared to SCAU141, the strain SCAU265 was isolated from the gorgonian coral Leptogorgia rigida. Morphological characteristics and genetic analysis based on internal transcribed spacer (ITS) rRNA sequences (accession number OR122480 in GenBank) were used to compare the strain with Aspergillus sp. 37a1 (accession number MN912600) in GenBank, revealing a high similarity of 99.82%. Based on this analysis, the strain SCAU265 was identified as Aspergillus sp..

  1. Despite the extensive research on secondary metabolites, there has been limited investigation into biological extracellular  polysaccharides derived from marine Aspergillus species. Some notable examples include the  exopolysaccharide AVP-141 from marine fungus Aspergillus versicolor SCAU141 [13], the antioxidant extracellular polysaccharide YSS from A. terreus [14], the exopolysaccharide AWP  isolated from a marine coral-associated A. versicolor [15], and an antioxidant exopolysaccharide  isolated from a marine-derived Aspergillus sp. Y1 [16].

 part of the answer

This finding aligns with the immune activity demonstrated by an other  marine fungal polysaccharide, AVP141-A, which was previously reported [13]. Moreover, ASP-1  significantly enhanced the production of TNF-α and IL-6 in RAW264.7 macrophages, indicating its  capacity to promote the activation of M1 immune cells and thereby mediate the inflammatory  response [41].

Response: Thank you very much for your positive comments. Your thoughtfulness is worth learning from you.

**********************************

All revised contents in text were highlighted in red in the revised manuscript.

Thank you very much for your consideration of our manuscript for potential publication. We look forward to hearing from you soon.

Best Regards.

Sincerely yours,

Dr. Xiaoyong Zhang

University Joint Laboratory of Guangdong Province, Hong Kong and Macao Region on Marine Bioresource Conservation and Exploitation, College of Marine Sciences, South China Agricultural University, Guangzhou 510642, China

E-mail: zhangxiaoyong@scau.edu.cn

Reviewer 3 Report

The manuscript describes the structure of a new exopolysaccharide isolated from a marine Aspergillus sp. strain. Data on its immunomodulatory effects are also given; this EPS can stimulate the immune response of cultured macrophage cells.

This manuscript needs many improvements. Material and methods section is not clear enough and several points are missing. The figures are not readable and not all in English, and the statistical analysis of metabolomics (PCA, OPLS-DA) is not given. Some information on the value of new compounds able to stimulate (or inhibit) the immune system and on the structure-activity relationships is also missing, including osidic composition, structure and molecular weight.

The activity section must be carefully considered; it raises several issues. Indeed, LPS is used as a positive control of the immune system activation; the conclusion is then the EPS can stimulate the inflammatory response at high concentration; is it worth testing the capacity of the immune system to fight against infection by Aspergillus sp. SCAU265; the question is important especially because a 400 µg/mL EPS concentration is needed? The activity of EPS in the presence of LPS was not studied, why? How do the authors plan to use EPS for immunomodulation? The rational behind this activity detection is not described.

I am not an expert in structural analysis, and cannot review this part.

The reference number in the main text do not correspond to the final list ; please check carefully.

Detailed comments :

Line 14 : Suppress this sentence, « crucial role » can be criticized

Line 22 : Glcp etc, as T-Glcp etc should be defined.

Introduction : a paragraph presenting immune system, especially macrophage, cytokine etc… roles, and immunomodulatory activity of EPS is lacking.

Lines 74-76 : Better than the technics would the feature analysed, such as molecular weight, glycosidic bonds, osidic composition etc…

Lines 86-89 : It is strange to identify the genus by using only one strain.

Line 95 and 101 : r/min ? maybe rpm should be better. r is not defined

Lines 97-99 : what become fungal cells in this process ? It seems they are still present in the concentrate. How is the concentration performed ? what is the system for reduced pressure? use of a membrane ? Which porosity ? what is the final volume ?

Line 103 : Ref 20 for the Sevage method ? Please cite the original article (Sevage et al ?)

Lines 105-107 : How is the elution perfomed ? what is the detection method ?

Line 110 : what is tandem gel ?

Line 120 : is it FTIR spectrum ?

Line 122 : Is there an hydrolysis step ? How is it performed ?

Section 2.5 : Please change the title, the purpose is not to analyse the methylation of the polysaccharide but the glycosidic bonds between residues.

Line 130 : the http address links to the University of Georgia. What is the database used ? How was it used ?

Line 131 : what is the purpose of NMR analysis ?

Line 140 : please add « macrophage cell ». What is this line specificity ? which provider ?

Lines 144-149 : What is the medium volume ? The ASP-A volume added ?

Lines 150-155 : « along with » is not clear, LPS seems to be mixed with EPS.

Lines 160 : what is the gel in the column ?

Line 169 : Please replacxe « plots » by « analyses ».

Line 170 : add an s to software. The reference 27 do not stand for any software description. You should here precise the right articles and software versions

Lines 174-175 : The p-value shows the likelihood of your data occurring under the null hypothesis. P-values help determine statistical significance but not the significance intensity. You cannot write « extremely ». It is only significant at pvalue<0.005 or pvalue <0.01.

You use both p and p-value. Thanks for homogenizing

Line 178 : 25L, in the methods section, you talk about 500 mL. What is the concentrate volume ? the volume after precipitation ?

Line 182 : DEAE-52 is not indicated in the methods section while G-100 is indicated and not G-75.

Figure 1 : enlarge all and use a good resolution. A and B, the detection is 490 nm, why ? it is not indicated previously. C : you indicate (490nm) while abscise is the wavelength.

1A : where is the EPS ? D : where are the standards used for the moleulcar weight determination ?

Fig1 caption : replace curves by chromatograms. What is IC ?

Fig2 : we can see nothing. It is not English.

Please add the structure in supplemental data in the core manuscript.

It is not true that the EPS backbone is Glc since Man and Gal are also encoutered (Lines 20, 193, 205, 339, 390). You should also discuss the particular structure with the square shape.

Fig3 A : activity or viability ?

Lines 268 and further : please improve the description of the choice of the indicators (either here or in the introduction as previously advised)

Line 290 : discuss the concentration levels. Has this EPS an interesting effect considering the concentrations ?

Line 295 : you should state you talk now of metabolomics of macrophages. Pvalue is the basis of your statistical analysis, but what about the fold-change, is a fold-change of 1.5 really confident ? Usually, we only keep fold change above 2 or below -2. PCA, OPLS-DA should be given and analysed because described in the material and methods.

In this paragaph, why do you only discuss on the amino-acids ? The table in the supplemental data also indicates some other compounds significantly differentially expressed. This table should also be improved by indicating the type of the compound (aminoacide etc…). Please order them by type, alphabetical… in lines 306-315, we do not know which ones are up or down regulated.

Discussion : must be improved (see above)

Author Response

Dear reviewer (3),

Journal of Fungi,

On behalf of my co-authors, I would like to submit the revised manuscript entitled “A new exopolysaccharide of marine coral-associated Aspergillus sp. SCAU265: Structural characterization and immunomodulatory activity” (jof-2615601 to Journal of Fungi.

The authors are grateful to the reviewers for their valuable comments and time, and all comments are responded carefully point by point.

**********************************

  1. The manuscript describes the structure of a new exopolysaccharide isolated from a marine Aspergillus sp. strain. Data on its immunomodulatory effects are also given; this EPS can stimulate the immune response of cultured macrophage cells.

(1)This manuscript needs many improvements. Material and methods section is not clear enough and several points are missing. (2) The figures are not readable and not all in English, and the statistical analysis of metabolomics (PCA, OPLS-DA) is not given. (3) Some information on the value of new compounds able to stimulate (or inhibit) the immune system and on the structure-activity relationships is also missing, including osidic composition, structure and molecular weight.

Response: (1) Material and methods section had been improved in the revised manuscript, please check the responses to Questions 9~28, and check the revised manuscript.

(2) Figures 1, 2 and 5 had been revised; please check the revised Figures 1, 2 and 5. And the statistical analysis of metabolomics (PCA, OPLS-DA) had been added into the revised manuscript. Please check the response to Question 36 (2).

(3)The following paragraphs and references had been added into the revised manuscript.

Moreover, at a concentration of 400 μg/mL, ASP-1 could significantly enhance the production of TNF-α and IL-6 in RAW264.7 macrophages, which was consistent with a previous relevant report [45]. Tian et al. obtained an exopolysaccharide EPS1 from fungal isolate Paecilomyces cicadae TJJ1213 and found that EPS1 (with the concentration of 400 μg/mL) could significantly up-regulate the mRNA expression of TNF-α and IL-6 in macrophage RAW 264.7 cells [45].

Although the corresponding relationship was not clearly elucidated, the monosaccharide composition, structure and molecular weight of ASP-1 might be related to its immunological activity. Compared to β-glucans, less attention had been paid to α-glucans on the immunological activity, but recently many exopolysaccharides with α-glucans had been reported to exhibit immunological activity, such as ASP-1 (this study) and CEPSN-1from Auricularia auricula-judae [46]. Similar to exopolysaccharide CEPSN-1, polysaccharide ASP-1 was also mainly composed of glucose, mannose and galactose, but its immune activity (action concentration and molecular weight were 400 μg/mL and 36.07 kDa, respectively) was lower than CEPSN-1 (action concentration and molecular weight were 200 μg/mL and 4.60 kDa, respectively); which was similar to previous reports supporting that there was a negative correlation between molecular weight and biological activity [46, 47].

[45] Tian, J.; Tang, C.; Wang, X.; Zhang, X.; Xiao, L.; Li, W. Supramolecular structure features and immunomodulatory effects of exopolysaccharide from Paecilomyces cicadae TJJ1213 in RAW264.7 cells through NF-κB/MAPK signaling pathways. International Journal of Biological Macromolecules 2022, 207, 464–474.

[46] Zhang, Y.; Zeng, Y.; Men, Y.; Zhang, J.; Liu H.; Sun, Y. Structural characterization and immunomodulatory activity of exopolysaccharides from submerged culture of Auricularia auricula-judae. International Journal of Biological Macromolecules 2018, 115, 978–984.

[47] Li, J.; Chi, Z.; Yu, L.; Jiang, F.; Liu, C. Sulfated modification, characterization, and antioxidant and moisture absorption/retention activities of a soluble neutral polysaccharide from Enteromorpha prolifera, International Journal of Biological Macromolecules 2017, 105, 1544–1553.

  1. The activity section must be carefully considered; it raises several issues. Indeed, LPS is used as a positive control of the immune system activation; (1) the conclusion is then the EPS can stimulate the inflammatory response at high concentration; is it worth testing the capacity of the immune system to fight against infection by Aspergillus sp. SCAU265; the question is important especially because a 400 µg/mL EPS concentration is needed? (2) The activity of EPS in the presence of LPS was not studied, why? (3) How do the authors plan to use EPS for immunomodulation? The rational behind this activity detection is not described.

Response: (1) This is a very good suggest, and we will test the capacity of the immune system to fight against infection by Aspergillus sp. SCAU265 according to your suggest. In this manuscript, we mainly focused on the structure and immune activity of the EPS (ASP-1).

When it comes to active concentration, many extracellular polysaccharides exhibited a good immune activity at the concentrations ranging from 200 to 400 µg/mL. For examples, EPS1 from fungal isolate Paecilomyces cicadae TJJ1213, 400 μg/mL (Tian, et al. International Journal of Biological Macromolecules 2022, 207, 464–474); R-17-EPS from Lactobacillus pentosus, 400 μg/mL (You et al., International Journal of Biological Macromolecules 2020, 158, 408–419); AVP141-A from marine-derived fungus Aspergillus versicolor SCAU141, 200 μg/mL (Wu et al., International Journal of Biological Macromolecules 2023, 227, 329–339).

(2) As you said, LPS was used as a positive control of the immune system activation in this study. Therefore, we only treated it as a comparative object and did not test the immune activity of EPS (ASP-1) in the presence of LPS. In many previous studies, the activity of EPS in the presence of LPS was not evaluated. For examples, Tian, et al. International Journal of Biological Macromolecules 2022, 207, 464–474; You et al., International Journal of Biological Macromolecules 2020, 158, 408–419; Wu et al., International Journal of Biological Macromolecules 2023, 227, 329–339.

(3) In fact, what we want to express is that the exopolysaccharide (ASP-1) has immune activity, rather than a potential immune modulator. Therefore, this sentence was modified as follows.

These findings provide fundamental scientific data for further research on its accurate molecular mechanism of immunomodulatory activity.

  1. I am not an expert in structural analysis, and cannot review this part.

Response: Regarding the structure, two other reviewers provided some comments, and I had also responded to all comments one by one. Please check the revised manuscript.

  1. The reference number in the main text do not correspond to the final list; please check carefully.

Response: The reference number in the main text and the correct citations had been revised, please check the responses to Questions 12, 13 and 25.

Detailed comments:

  1. Line 14 : Suppress this sentence, « crucial role » can be criticized

Response: The sentence had been removed from the revised manuscript, and the following sentence had been added into the revised manuscript.

Recent studies have found that many of marine microbial polysaccharides exhibit distinct immune activity.

  1. Line 22 : Glcp etc, as T-Glcp etc should be defined.

Response: The sentences had been modified as follows.

Structural analysis revealed that ASP-1 had an average molecular weight of 36.07 kDa and was mainly composed of (1→4)-linked α-D-glucopyranosyl residues, along with highly branched heteropolysaccharide regions containing 1,4,6-glucopyranosyl, 1,3,4-glucopyranosyl, 1,4,6- galactopyranosyl, T-glucopyranosyl, T-mannopyranosyl, and T-galactopyranosyl residues.

  1. Introduction : a paragraph presenting immune system, especially macrophage, cytokine etc… roles, and immunomodulatory activity of EPS is lacking.

Response: The immunomodulatory activity of EPS (and the reference) was added into the revised manuscript, which was as follows.

Moreover, an extracellular polysaccharide (EAPS) derived from Rhodotorula sp. RY1801 exhibited a very strong immunomodulatory effect on nematodes [6].

[6] Wang, Z.; Zhao, Y.; Jiang, Y.; Chu, W. Prebiotic, antioxidant, and immunomodulatory properties of acidic exopolysaccharide from marine Rhodotorula RY1801. Frontier in Nutrition 2021, 8, 710688.

  1. Lines 74-76 : Better than the technics would the feature analysed, such as molecular weight, glycosidic bonds, osidic composition etc…

Response: These sentences were modified in the revised manuscript, which was as follows.

And the molecular weight, glycosidic bonds and monosaccharide compositions of ASP-1 were investigated by Fourier transform-infrared spectroscopy (FT-IR), gas chromatography-mass spectrometry (GC–MS), and nuclear magnetic resonance (NMR) spectroscopy, etc..

  1. Lines 86-89 : It is strange to identify the genus by using only one strain.

Response: In fact, the strain (a known fungal genus) had been identified by combining ITS sequences (molecular) with morphological observation (phenotypic) methods. In the process of identification of ITS sequence (molecular), the ITS sequences of the strain was compared on Genbank (https://www.ncbi.nlm.nih.gov), and the first three strains with the highest similarity were used as a reference. Only one with the highest similarity 99.82% (Aspergillus sp. 37a1, accession number MN912600) of them is listed in the manuscript.

Similar identification method have been recognized by many previous studies, For examples, Liao et al., J. Fungi 2023, 9, 613; Wu et al., International Journal of Biological Macromolecules 2023, 227: 329-339.

  1. Line 95 and 101 : r/min ? maybe rpm should be better. r is not defined

Response: Changed “r/min” to “rpm”.

  1. Lines 97-99 : what become fungal cells in this process ? It seems they are still present in the concentrate. How is the concentration performed ? what is the system for reduced pressure? use of a membrane ? Which porosity ? what is the final volume ?

Response: In fact, we have already removed fungal cells (with cheesecloth and Buchner funnel) before extracting extracellular polysaccharides. Perhaps our description is not very clear, but now the accurate description is as follows.

After a 7-day incubation period, the entire fermentation broth (approximately 25 L) was filtered through cheesecloth and Buchner funnel (with filter paper, pore size 30-50 μm) to remove the fungal mycelia. The filtered broth was then concentrated to 5 L under reduced pressure at a temperature of 55 °C.

So in the subsequent extraction process, there are no fungal cells present, only the fermentation broth is used for subsequent extraction.

  1. Line 103 : Ref 20 for the Sevage method ? Please cite the original article (Sevage et al ?)

Response: The Ref 20 for the Sevage method had been modified by the original article as follows. Staub A.M. Removal of proteins-Sevag method. Methods in carbohydrate chemistry, Academic Press: New York, USA, 1956.

  1. Lines 105-107 : How is the elution perfomed ? what is the detection method ?

Response: The crude polysaccharide ASP was fractionated on a Diethylaminoethyl-Sepharose-52 (DEAE-52) Fast Flow column (2.6 cm × 70 cm) by eluting with a step gradient of 0–1.2 mol/L NaCl solution at a flow rate of 1.0 mL/min. Eluate was collected by auto-fraction collector (10 mL/tube). Each tube was tested for carbohydrate content by the classic phenol-sulfuric acid method with the detection wavelength of 490 nm [21]. According to the profile of the gradient elution, the crude polysaccharide could be isolated two fractions. The fractions eluted with ultrapure water were pooled, dialyzed and loaded onto a Sephadex G-75 column (2.6 cm × 60 cm), and eluted with ultrapure water at a flow rate of 0.4 mL/min.

These sentences and the following references were added into the revised manuscript.

[21] Dubois, M.; Gilles, K.A.; Hamilton, J.K.; Rebers, P.A.; Smith, F. Colorimetric method for determination of sugars and related substances. Analytical Chemistry 1956, 28, 350–356.

  1. Line 110 : what is tandem gel ?

Response: This usage of “tandem gel” is incorrect, and the correct one should be “tandem column”. Changed “tandem gel column” to “tandem column” in the revised manuscript.

  1. Line 120 : is it FTIR spectrum ?

Response: Yes, it is FTIR spectrum. Changed “an FT-IR spectrometer” to “a FTIR spectrum” in the revised manuscript.

  1. Line 122 : Is there an hydrolysis step ? How is it performed ?

Response: Yes, there is a hydrolysis step. Weigh 5 mg of the sample (ASP-1) accurately and place it in an ampoule bottle. Add 2mL of 3 mol/L Trifluoroacetic acid (TFA) solution and hydrolyze at 120 ℃ for 3 hours. And the sentence (Line 122) was modified as follows.

After hydrolyzed with 3 mol/L Trifluoroacetic acid (TFA) solution at 120 ℃ for 3 hours, the monosaccharide compositions of ASP-1 were determined using ion chromatography (ICS5000, ThermoFisher, USA) following the method described in a previous study [19].

  1. Section 2.5 : Please change the title, the purpose is not to analyse the methylation of the polysaccharide but the glycosidic bonds between residues.

Response: The title was modified as “Analysis of the glycosidic bonds between residues of ASP-1”. And the sentence (Line 126) was modified as “Analysis of the glycosidic bonds between residues of ASP-1 was conducted following Hakomori's method [25]”.

  1. Line 130 : the http address links to the University of Georgia. What is the database used ? How was it used ?

Response: After opening this website (University of Georgia), find "resources" and pull down to find "GC-EIMSPMAAs" for data analysis. As shown in the following figure.

  1. Line 131 : what is the purpose of NMR analysis ?

Response: NMR analysis was used to identify the molecular structure of ASP-1, for example, the number of H and C, the correlation between H and C, and the correlation between H and H, etc..

  1. Line 140 : please add « macrophage cell ». What is this line specificity ? which provider ?

Response: Macrophage cell was added into the revised manuscript, and the sentence was modified as follows.

The macrophage RAW 264.7 cell line (Procell, China) was utilized to assess the immunomodulatory activity of ASP-1.

The macrophage RAW 264.7 cell line was provided by Procell, China.

And the specificity of the macrophage RAW 264.7 cell line described as follows.

The RAW264.7 cell line is derived from male mice and is a blood derived macrophage. RAW264.7 cells have strong phagocytic ability. After phagocytosis of antigens, the cells release chemokines, promoting differentiation, extending pseudopodia, and enhancing climbing ability.

The RAW264.7 cell line primarily recognize receptors and pathogen activation related molecules through their expression patterns, thereby engulfing pathogens. Autophagy is a process of maintaining homeostasis in the body, which can engulf its own cytoplasmic proteins or organelles and envelop them into vesicles, and fuse with lysosomes to form autophagic lysosomes, degrading their encapsulated contents, thereby achieving the metabolic needs of the cell itself and the renewal of certain organelles. So when studying the regulation of immune response and inflammatory response processes, this cell line is often used.

  1. Lines 144-149 : What is the medium volume ? The ASP-A volume added ?

Response: The polysaccharide ASP-1 was dissolved in the medium to obtain a series of different concentration ASP-1 solutions (20, 10, 4, 2, 1, 0.5 and 0 mg/mL), and then mix them with the medium (including macrophage RAW 264.7 cells) according to the final concentration (0, 10, 50, 100, 200, 500, and 1000 μg/mL) of ASP-1 we want to set. The total volume is 100 μL in each well, so the ASP-1 volume (ASP-1 solutions with different concentrations) and medium volume (including macrophage RAW 264.7 cells) and were described in the following table.

Final Concentration (μg/mL)

0

10

50

100

200

500

1000

Concentration of ASP-1

0

0.5

1

2

4

10

20

The volume of ASP-1 solution (μL)

5

5

5

5

5

5

5

The volume of Medium (μL)

(including macrophage RAW 264.7 cells)

95

95

95

95

95

95

95

  1. Lines 150-155 : « along with » is not clear, LPS seems to be mixed with EPS.

Response: The sentences were modified as follows.

Subsequently, the cells were treated with ASP-1 at different concentrations (100, 200 and 400 μg/mL) and lipopolysaccharide (LPS) at a concentration of 2.5 μg/mL, respectively, and then further incubated for 24 hours.

  1. Lines 160 : what is the gel in the column ?

Response: In the ultrahigh pressure liquid chromatography (UPLC) system, the column is C18 column.

The sentence was modified as follows.

An ultrahigh pressure liquid chromatography (UPLC) system equipped with a 2.1 mm × 150 mm × 1.8 μm C18 column was used for sample separation and analysis.

  1. Line 169 : Please replacxe « plots » by « analyses ».

Response: Changed “plots” to “analyses”. Please check the revised manuscript.

  1. Line 170 : add an s to software. The reference 27 do not stand for any software description. You should here precise the right articles and software versions.

Response: Added s to software. Two references (including Ref 27) were modified, and the right articles and software versions were corrected in the revised manuscript.

The sentence and references were modified as follows.

Analyses were generated using R, MetaboAnalyst (Version 4.0) [28] and KEGG Mapper (Version 5, released in July 2021) softwares [29].

[28] Xia, J.; Psychogios, N.; Young, N.; Wishart, D.S. MetaboAnalyst: a web server for metabolomic data analysis and interpretation. Nucleic Acids Research, 2009, 37, W652–W660.

[29] Kanehisa, M.; Sato, Y.; Kawashima, M. KEGG mapping tools for uncovering hidden features in biological data. Protein Science 2021, 31, 47–53.

  1. Lines 174-175 : The p-value shows the likelihood of your data occurring under the null hypothesis. P-values help determine statistical significance but not the significance intensity. You cannot write « extremely ». It is only significant at p value<0.005 or pvalue <0.01. You use both p and p-value. Thanks for homogenizing.

Response: The sentence was modified as follows.

A significance level of p-value < 0.05 or p-value < 0.01 was considered statistically significant [26].

And p and p-value were homogenized with p-value. Please check the revised manuscript.

  1. Line 178 : 25L, in the methods section, you talk about 500 mL. What is the concentrate volume ? the volume after precipitation ?

Response: The 500 mL mentioned in the manuscript referred to the volume of the Erlenmeyer flask (containing 120 mL of culture medium for fermentation) we use for fermentation. A total of 210 Erlenmeyer flasks were used for fermentation and 25 L of fermentation broth was obtained. After concentration, the concentrate volume of fermentation broth was approximately 5 L. After precipitated, the volume of precipitation was approximately 2.5 L.

  1. Line 182 : DEAE-52 is not indicated in the methods section while G-100 is indicated and not G-75.

Response: The sentence (in the methods section) was modified as follows.

The crude polysaccharide ASP was fractionated on a Diethylaminoethyl-Sepharose-52 (DEAE-52) Fast Flow column (2.6 cm × 70 cm) by eluting with a step gradient of 0–1.2 mol/L NaCl solution at a flow rate of 1.0 mL/min.

The fractions eluted with ultrapure water were pooled, dialyzed and loaded onto a Sephadex G-75 column (2.6 cm × 60 cm), and eluted with ultrapure water at a flow rate of 0.4 mL/min.

  1. Figure 1 : (1) enlarge all and use a good resolution. (2)A and B, the detection is 490 nm, why ? it is not indicated previously. (3) C : you indicate (490nm) while abscise is the wavelength.

(4) 1A : where is the EPS ? D : where are the standards used for the moleulcar weight determination ?

(5) Fig1 caption : replace curves by chromatograms. What is IC ?

Response: (1) The figure 1 had been enlarge with a 300 dpi resolution. Please check the following revised Figure 1.

Fig. 1 (revised) ASP-1 elution chromatogram on DEAE-52 Fast Flow (a) and Sephadex G-75 (b), UV spectrum of ASP-1 (c), monosaccharides of ASP-1 by Ion Chromatogram (d), HPGPC chromatogram of ASP-1 (e) and FT-IR spectrum of ASP-1 (f).

(2) Under the action of concentrated sulfuric acid, the dehydration of exopolysaccharides can condense with phenol to form an orange red compound. In the range of 10-100 mg, its color depth is directly proportional to the sugar content, and there is a maximum absorption peak at the wavelength of approximately 490 nm. Therefore, the detection is 490 nm (classic phenol-sulfuric acid method).

The detection method related to the Figures A and B (added “with the detection wavelength of 490 nm”) has been modified as follows.

Each tube was tested for carbohydrate content by the classic phenol-sulfuric acid method with the detection wavelength of 490 nm [21].

(3) You are right. The 490 nm were deleted from the Fig. 1C. Please check the above Fig 1(revised).

(4) The EPS was the first and high peak, and the EPS (ASP) had been marked in the Fig.1c. Please check the above Fig.1c.

The standards used for the moleulcar weight determination was shown the follow Figure (a), and the detection of monosaccharide of EPS (ASP-1) was shown the follow Figure (b). Considering that there are too many images in Fig. 1, the standards used for the moleulcar weight determination (the follow Figure (a) is not included in the revised manuscript.

(5) The caption of Fig.1 had been modified as “Fig. 1. ASP-1 elution chromatogram on DEAE-52 Fast Flow (a) and Sephadex G-75 (b), UV spectrum of ASP-1 (c), monosaccharides of ASP-1 by Ion Chromatogram (d), HPGPC chromatogram of ASP-1 (e) and FT-IR spectrum of ASP-1 (f)”.

IC is incorrect, it is ion. So “IC Chromatogram” was changed to “Ion Chromatogram”.

  1. Fig2 : we can see nothing. It is not English.

Response: Please forgive our negligence. Now I had made some modifications. Please check the following revised Fig. 2.

Fig. 2 (revised) NMR analysis of ASP-1; 1H NMR (a); 13C NMR (b); 1H-1H COSY (c); HSQC (d); HMBC (e); the chemical structure of ASP-1 (f), Residue A: T-α-Glcp, Residue B: T-β-Manp, Residue C: T-α-Galp, Residue D: 1,2-α-Manp, Residue E: 1,3-α-Glcp, Residue F: 1,4-α-Glcp, Residue G: 1,6-β-Glcp, Residue H: 1,3,4-α-Glcp, Residue I: 1,4,6-α-Glcp, Residue J: 1,4,6-α-Galp.

  1. Please add the structure in supplemental data in the core manuscript.

Response: The structure in supplemental data had been added into the revised manuscript, please check the above revised Fig. 2f.

  1. (1) It is not true that the EPS backbone is Glc since Man and Gal are also encoutered (Lines 20, 193, 205, 339, 390). (2)You should also discuss the particular structure with the square shape.

Response: (1) The backbone was removed from these sentences (Lines 20, 193, 205, 339, 390), and these sentence were modified as follows.

Line 20 (revised ) : Structural analysis revealed that ASP-1 has an average molecular weight of 36.07 kDa and is mainly composed of (1→4)-linked α-D-glucopyranosyl residues, along with highly branched heteropolysaccharide regions containing 1,4,6-glucopyranosyl, 1,3,4-glucopyranosyl, 1,4,6- galactopyranosyl, T-glucopyranosyl, T-mannopyranosyl, and T-galactopyranosyl residues.

Line193: This composition suggests that ASP-1 likely possesses a main component of glucose.

Line 205: This composition suggests that ASP-1 likely possesses a main component of glucose

Line 339:These findings suggest that ASP-1 may have a main structure consisting of (1→4)-linked α-D-Glcp residues, with highly branched regions containing 1,4,6-glucopyranosyl (Glcp) , 1,3,4-Glcp, 1,4,6-Galp, T-Glcp, T- mannopyranosyl (Manp), and T- galactopyranosyl (Galp) residues, thus forming a heteropolysaccharide structure.

Line 390:Its structural characterization revealed a mainly composed of (1 → 4)-linked α-D-Glcp residues, accompanied by highly branched heteropolysaccharide regions containing 1,4,6-Glcp, 1,3,4-Glcp, 1,4,6-Galp, T-Glcp, T-Manp, and T-Galp residues.

(1) The following sentences were added into the Part of Discussions.

Interestingly, four 1,4,6-α-Glcp residues of ASP-1 were connected together to form a square structure (Fig. 2f), which was relatively rare in marine fungal extracellular polysaccharides.

  1. Fig3 A : activity or viability ?

Response: The word of “activity” had been changed to “viability” in the revised Fig.3.

  1. Lines 268 and further : please improve the description of the choice of the indicators (either here or in the introduction as previously advised)

Response: The description of the choice of the indicators had been improved as follows.

During the immune response process, the production of active substances plays a vital role in immune regulation. Nitric oxide (NO) is one such active substance involved in immune regulation, and its release serves as an indicator of immune modulation. When assessing the immune regulatory potential of a substance, measuring NO release of RAW 264.7 macrophages can provide valuable insights. In this study, treatment with low concentrations of ASP-1 (100 and 200 μg/mL) did not significantly affect NO release compared to the blank control group. At a dose of 400 μg/mL, ASP-1 demonstrated a significant promotion of NO release (p-value < 0.001), but the NO release level was lower than that in the positive control group (2.5 μg/mL LPS) (Fig. 4a). The noteworthy increase in NO release suggested that ASP-1 possessed potent immune-stimulating effects on RAW 264.7 macrophages.

The release of cytokines TNF-α and IL-6 was assessed using ELISA kits. When RAW 264.7 macrophages were treated with ASP-1 at concentrations of 100, 200, and 400 μg/mL for 24 hours, the secretion of TNF-α increased significantly, presenting a clear dose-dependent pattern (Fig. 4b). Furthermore, at a concentration of 400 μg/mL, the secretion of TNF-α surpassed that of the positive control group treated with LPS (Fig. 4b) (p-value < 0.001). As for the effect of ASP-1 on IL-6 secretion, it was found that ASP-1 with high concentration (400 μg/mL) could significantly increase IL-6 secretion (p-value < 0.001); While the low concentrations (100 and 200μg/mL) of ASP-1 had no significant effect on IL-6 secretion (Fig. 4c).

  1. Line 290 : discuss the concentration levels. Has this EPS an interesting effect considering the concentrations ?

Response: The following sentences and reference were added into the Parts of Discussion and Reference.

Moreover, at a concentration of 400 μg/mL, ASP-1 could significantly enhance the production of TNF-α and IL-6 in RAW264.7 macrophages, which was consistent with a previous relevant report [43]. Tian et al. obtained an exopolysaccharide EPS1 from fungal isolate Paecilomyces cicadae TJJ1213 and found that EPS1 (with the concentration of 400 μg/mL) could significantly up-regulate the mRNA expression of TNF-α and IL-6 in macrophage RAW 264.7 cells [43].

[43] Tian, J.; Tang, C.; Wang, X.; Zhang, X.; Xiao, L.; Li, W. Supramolecular structure features and immunomodulatory effects of exopolysaccharide from Paecilomyces cicadae TJJ1213 in RAW264.7 cells through NF-κB/MAPK signaling pathways. International Journal of Biological Macromolecules 2022, 207, 464–474.

  1. Line 295 : (1) you should state you talk now of metabolomics of macrophages. (2) Pvalue is the basis of your statistical analysis, but what about the fold-change, is a fold-change of 1.5 really confident ? Usually, we only keep fold change above 2 or below -2. (3) PCA, OPLS-DA should be given and analysed because described in the material and methods.

Response: (1) The sentence was modified as follows. In the process of metabolomics analysis of macrophage RAW 264.7 cells treated with ASP-1, metabolites with p-values less than 0.05 are considered as differential metabolites, while metabolites with p-values between 0.05 and 0.1 are considered as metabolites with marginal significance.

(2) Your are right. The result of a fold-change of 1.5 had been removed from the manuscript, and other results had been added into the revised manuscript. Which were as follows.

The metabolomics of macrophage RAW 264.7 cells treated with ASP-1 and control groups were performed using UPLC and MS-ESI systems. A total of 4054 metabolites were obtained in treatment and control groups. The PCA highlighted the characteristic of differences in treatment and control groups. PC1 and PC2 explained 43.8% and 16.2% of variance in PCA score plot, respectively (Fig. s6), indicating significant differences of the metabolites among treatment and control groups. The result was also supported by OPLS-DA (Fig. s7). A total of 64 metabolites were identified as significantly different, with 28 upregulated and 36 downregulated metabolites. Among these significantly different metabolites, amino acids, peptides and analogues (such as citrulline, glutamic acid, aspartic acid and isoleucine, etc.) exhibited notable changes following ASP-1 treatment (Table s1, supplemental materials). The most prominently affected amino acid was citrulline, increased by 86.22-fold; while aspartic acid and isoleucine were significantly decreased to 0.42-fold and 0.34-fold, respectively. Furthermore, fatty acids and conjugates, and carbohydrates and carbohydrate conjugates also showed significant changes (Table s1, supplemental materials). For examples, itaconic acid, undecanoic acid, D-galactose and trans-1,2-cyclohexanediol were increased by 21.58-, 4.09-, 3.82- and 3.37-fold, respectively; while mannitol and sorbitol were decreased to 0.23- and 0.27-fold, respectively (Table s1, supplemental materials). In addition, other types of metabolites were relatively less distributed and had not been further analyzed in this study.

(3)The results of PCA and OPLS-DA had been described as follows.

The PCA highlighted the characteristic of differences in treatment and control groups. PC1 and PC2 explained 43.8% and 16.2% of variance in PCA score plot, respectively (Fig. s6), indicating significant differences of the metabolites among treatment and control groups. The result was also supported by OPLS-DA (Fig. s7).

Fig. s6 Principal Component Analysis (PCA) score plot of the macrophage RAW 264.7 cell samples from ASP-1 treatment and control groups in positive (a) and negative (b) ion modes. Con_1, 2 and 3 represent three parallel control samples, and AS_1, 2 and 3 represent three parallel treated samples with ASP-1.

Fig. s7 Orthogonal partial least-squares discriminant analysis (OPLS-DA) score plot of the macrophage RAW 264.7 cell samples from ASP-1 treatment and control groups in positive (a) and negative (b) ion modes. Con_1, 2 and 3 represent three parallel control samples, and AS_1, 2 and 3 represent three parallel treated samples with ASP-1.

  1. (1) In this paragraph, why do you only discuss on the amino-acids ? (2)The table in the supplemental data also indicates some other compounds significantly differentially expressed. This table should also be improved by indicating the type of the compound (aminoacide etc…). Please order them by type, alphabetical… (3) in lines 306-315, we do not know which ones are up or down regulated.

Response: (1) In differential analysis of metabolic products, there are many types of amino acids, and many pathways are related to the metabolism of amino acids in the process of Metabolomics analysis Among the top 20 significantly different metabolic pathways (Fig. 5), the majority exhibited significant differences in amino acid metabolism (Table s2, supplemental materials). Further analysis of the significant differential metabolites within each pathway highlighted the frequent enrichment of metabolites such as citrulline, glutamic acid, aspartic acid, and isoleucine. This indicates that changes in amino acid metabolism are the most prominent metabolic alterations in RAW264.7 macrophages treated with ASP-1. This result was also similar to a previous literature report (Wu et al., International Journal of Biological Macromolecules 2023, 227: 329-339). Additionally, there is a literature report on how amino acid metabolism regulates immune activity (Kelly B, Pearce EL. Amino assets: how amino acids support immunity. Cell Metabolism 2020, 32:154-175). All of the above result and literatures lead us to speculate that amino acid metabolism might be the potential mechanisms of immune regulatory of ASP-1. So, only amino-acids were discussed in this paragraph.

Of course, this is just a speculation, which may require more evidence to support our hypothesis. In future experiments, we will use various methods to verify our hypothesis.

 (2) This table had been improved by indicating the type of the compound (Class), please check the following table s1.

Table s1 Significant differences in metabolites between the ASP-1 treatment group and the control group

Metabolites

Class

VIP

FC

p-value

L-threo-3-Methylmalate

Alcohols and polyols

1.6485

8.66

< 0.0001

IMP

Alcohols and polyols

1.6359

2.2

0.0001

Phosphohydroxypyruvic acid

Alpha-keto acids and derivatives

1.4011

1.05

0.0327

Porphobilinogen

Amines

1.4694

0.73

0.0014

Citrulline

Amino acids, peptides, and analogues

1.5132

86.22

< 0.0001

N-Acetylglutamic acid

Amino acids, peptides, and analogues

1.6465

1.93

< 0.0001

L-Glutamic acid

Amino acids, peptides, and analogues

1.6408

1.59

< 0.0001

Dimethylglycine

Amino acids, peptides, and analogues

1.636

1.54

0.0001

Asymmetric dimethylarginine

Amino acids, peptides, and analogues

1.4012

0.75

0.0083

L-Asparagine

Amino acids, peptides, and analogues

1.5852

0.73

0.0022

L-Serine

Amino acids, peptides, and analogues

1.6091

0.72

0.0009

L-Aspartic acid

Amino acids, peptides, and analogues

1.6459

0.42

< 0.0001

Creatinine

Amino acids, peptides, and analogues

1.355

0.36

0.0162

L-Isoleucine

Amino acids, peptides, and analogues

1.2413

0.34

0.0465

(2E)-Decenoyl-ACP

Amino acids, peptides, and analogues

1.4647

0.2

0.0017

Metoclopramide

Aminophenyl ethers

1.2462

0.68

0.0455

Dehydroepiandrosterone

Androstane steroids

1.2769

0.09

0.0352

Salicylic acid

Benzoic acids and derivatives

1.5322

0.2

0.036

Phthalic acid

Benzoic acids and derivatives

1.3903

0.09

0.0352

D-Galactose

Carbohydrates and carbohydrate conjugates

1.4425

3.83

0.0234

trans-1,2-Cyclohexanediol

Carbohydrates and carbohydrate conjugates

1.5167

3.37

0.0092

N-Acetyl-a-neuraminic acid

Carbohydrates and carbohydrate conjugates

1.5913

0.44

0.0018

Sorbitol

Carbohydrates and carbohydrate conjugates

1.2954

0.27

0.0307

Mannitol

Carbohydrates and carbohydrate conjugates

1.3061

0.23

0.0266

Cholesterol

Cholestane steroids

1.4685

0.02

0.0014

Cyclic GMP

Cyclic purine nucleotides

1.2344

2.76

0.0472

Succinic acid

Dicarboxylic acids and derivatives

1.6484

4.73

< 0.0001

19(S)-HETE

Eicosanoids

1.3466

1.29

0.0475

Propionylcarnitine

Fatty acid esters

1.4458

0.73

0.0031

Itaconic acid

Fatty acids and conjugates

1.6485

21.58

< 0.0001

Undecanoic acid

Fatty acids and conjugates

1.3036

4.09

0.0276

Palmitic acid

Fatty acids and conjugates

1.3456

3.23

0.0464

7,8-Diaminononanoate

Fatty acids and conjugates

1.5015

2.19

0.0001

Docosapentaenoic acid (22n-3)

Fatty acids and conjugates

1.5057

0.85

0.0107

5-Guanidino-3-methyl-2-oxopentanoate

Fatty acids and conjugates

1.2722

0.78

0.0365

Palmitoleic acid

Fatty acids and conjugates

1.3873

0.53

0.0106

Glycerophosphocholine

Glycerophosphocholines

1.4362

2.18

0.004

Gamma-Linolenic acid

Lineolic acids and derivatives

1.4481

2.55

0.003

Oxoadipic acid

Medium-chain keto acids and derivatives

1.6461

4.38

< 0.0001

3-Carbamoyl-2-phenylpropionaldehyde

Phenylacetaldehydes

1.394

1.34

0.0095

O-Phosphoethanolamine

Phosphate esters

1.3125

0.34

0.0256

Pantothenic acid

Polyols

1.597

1.25

0.015

Pterin

Pterins and derivatives

1.5008

2.61

0.0001

1-Methyladenosine

Purine nucleosides

1.5033

0.34

0.0001

Xanthylic acid

Purine ribonucleotides

1.431

0.56

0.0244

Xanthine

Purines and purine derivatives

1.5746

0.85

0.0031

Pyridoxal 5'-phosphate

Pyridine carboxaldehydes

1.4346

0.61

0.0043

Quinolinic acid

Pyridinecarboxylic acids and derivatives

1.4899

2.63

0.0004

Pyridoxine

Pyridoxines

1.3949

0.63

0.0093

Uridine

Pyrimidine nucleosides

1.6407

1.41

< 0.0001

Uridine diphosphate-N-acetylglucosamine

Pyrimidine nucleotide sugars

1.472

0.33

0.0173

Cytosine

Pyrimidines and pyrimidine derivatives

1.4032

1.41

0.0079

Cytidine

Pyrimidines and pyrimidine derivatives

1.44

1.4

0.0036

Aprobarbital

Pyrimidines and pyrimidine derivatives

1.6484

0.25

< 0.0001

L-Carnitine

Quaternary ammonium salts

1.4704

0.54

0.0013

Fumaric acid

Short-chain keto acids and derivatives

1.6191

0.81

0.0005

(2R)-2-Hydroxy-3-(phosphonatooxy)propanoate

Sugar acids and derivatives

1.5688

1.2

0.0035

Citric acid

Tricarboxylic acids and derivatives

1.6378

1.92

0.0001

Phenyl acetate

others

1.5735

0.34

0.0029

2-Deoxystreptamine

others

1.2444

0.75

0.0442

2-Acetamidofluorene

others

1.3594

0.68

0.0155

2-Iminobutanoate

others

1.2298

0.67

0.049

NAD

others

1.3246

0.3

0.0226

VIP: Variable importance in projection, FC: Fold Change

 (3) The number of up or down metabolites in the top 20 enrichment pathways was shown in Revised Fig. 5, and these metabolites were shown in table s2.

                                  The number of significant differential metabolites

Fig. 5 (revised) Histogram of top 20 enrichment of differential metabolic pathways

Table s2 Up- or down-regulated metabolites in the top 20 enrichment pathways (The ones marked in red are amino acids)

Pathway (Top 20 enrichment)

Up metabolites

Down  metabolites

Central carbon metabolism in cancer

L-Glutamic acid; Succinic acid; L-Aspartic acid; L-Serine

Fumaric acid; L-Asparagine; Citric acid; (2R)-2-Hydroxy-3-(phosphonatooxy)propanoate; L-Isoleucine

Biosynthesis of plant secondary metabolites

L-Glutamic acid; Succinic acid; L-Aspartic acid; L-Serine; Fumaric acid; IMP; L-Asparagineacid

Citric acid; (2R)-2-Hydroxy-3-(phosphonatooxy) propanoate; Palmitic acid; Xanthine; L-Isoleucine; Xanthylic acid; Quinolinic acid

Biosynthesis of alkaloids derived from histidine and purine

Succinic acid; Fumaric acid; IMP; Citric acid

(2R)-2-Hydroxy-3-(phosphonatooxy)propanoate; Xanthine; Xanthylic acid

Biosynthesis of amino acids

L-Glutamic acid; L-Aspartic acid; L-Serine; L-Asparagine; Citric acid; (2R)-2-Hydroxy-3-(phosphonatooxy) propanoate; Oxoadipic acid

Citrulline; L-Isoleucine; N-Acetylglutamic acid; Phosphohydroxypyruvic acid

Alanine, aspartate and glutamate metabolism

L-Glutamic acid; Succinic acid; L-Aspartic acid

Fumaric acid; L-Asparagine; Citric acid

Biosynthesis of alkaloids derived from ornithine, lysine and nicotinic acid

L-Glutamic acid; Succinic acid; L-Aspartic acid; Fumaric acid; Citric acid

(2R)-2-Hydroxy-3-(phosphonatooxy) propanoate; L-Isoleucine; Quinolinic acid

Arginine biosynthesis

L-Glutamic acid; L-Aspartic acid; Fumaric acid

Citrulline; N-Acetylglutamic acid

Biosynthesis of cofactors

NAD; Pyridoxal 5'-phosphate; L-Glutamic acid; L-Aspartic acid; L-Serine; IMP; Citric acid; (2R)-2-Hydroxy-3-(phosphonatooxy) propanoate

Pyridoxine; Oxoadipic acid; Pantothenic acid; Porphobilinogen; 7,8-Diaminononanoate; Quinolinic acid

Biosynthesis of plant hormones

Succinic acid; L-Aspartic acid; Fumaric acid; IMP

Citric acid; (2R)-2-Hydroxy-3-(phosphonatooxy)propanoate; Salicylic acid

ABC transporters

L-Glutamic acid; L-Aspartic acid; L-Serine

Uridine; Mannitol; L-Isoleucine; Cytidine; Sorbitol; Phthalic acid

Carbon metabolism

L-Glutamic acid; Succinic acid; L-Aspartic acid; L-Serine; Fumaric acid

Citric acid; (2R)-2-Hydroxy-3-(phosphonatooxy) propanoate; Phosphohydroxypyruvic acid

Biosynthesis of various secondary metabolites - part 3

L-Glutamic acid; L-Aspartic acid; L-Serine; Citric acid

(2R)-2-Hydroxy-3-(phosphonatooxy) propanoate; Citrulline

Protein digestion and absorption

L-Glutamic acid

L-Aspartic acid; L-Serine; L-Asparagine; L-Isoleucine

Biosynthesis of alkaloids derived from terpenoid and polyketide

Succinic acid; Fumaric acid; Citric acid

Cholesterol; (2R)-2-Hydroxy-3-(phosphonatooxy) propanoate

Glucagon signaling pathway

Succinic acid; Fumaric acid; Citric acid

(2R)-2-Hydroxy-3-(phosphonatooxy) propanoate

Glycine, serine and threonine metabolism

L-Aspartic acid; L-Serine; (2R)-2-Hydroxy-3-(phosphonatooxy) propanoate; Dimethylglycine; Phosphohydroxypyruvic acid

Dimethylglycine; Phosphohydroxypyruvic acid

Aminoacyl-tRNA biosynthesis

L-Glutamic acid

L-Aspartic acid; L-Serine; L-Asparagine; L-Isoleucine

Two-component system

L-Glutamic acid; Succinic acid; L-Aspartic acid

Fumaric acid; Citric acid

Mineral absorption

L-Serine

D-Galactose; L-Asparagine; L-Isoleucine

Nicotinate and nicotinamide metabolism

NAD; Succinic acid

L-Aspartic acid; Fumaric acid; Quinolinic acid

  1. Discussion : must be improved (see above)

Response: The Part of Discussion had been modified in the revised manuscript, please check the revised manuscript.

**********************************

All revised contents in text were highlighted in red in the revised manuscript.

Thank you very much for your consideration of our manuscript for potential publication. We look forward to hearing from you soon.

Best Regards.

Sincerely yours,

Dr. Xiaoyong Zhang

University Joint Laboratory of Guangdong Province, Hong Kong and Macao Region on Marine Bioresource Conservation and Exploitation, College of Marine Sciences, South China Agricultural University, Guangzhou 510642, China

E-mail: zhangxiaoyong@scau.edu.cn

Round 2

Reviewer 1 Report

The authors have sufficiently addressed the comments and concerns of this reviewer.  

Author Response

Dear reviewer (1),

Journal of Fungi,

On behalf of my co-authors, I would like to submit the revised manuscript entitled “A new exopolysaccharide of marine coral-associated Aspergillus sp. SCAU265: Structural characterization and immunomodulatory activity” (jof-2615601 to Journal of Fungi.

The authors are grateful to the reviewers for their valuable comments and time, and all comments are responded carefully point by point.

**********************************

The authors have sufficiently addressed the comments and concerns of this reviewer.  

Response: The authors are grateful to you for your valuable comments and time.

**********************************

All revised contents in text were highlighted in red in the revised manuscript.

Thank you very much for your consideration of our manuscript for potential publication. We look forward to hearing from you soon.

Best Regards.

Sincerely yours,

Dr. Xiaoyong Zhang

University Joint Laboratory of Guangdong Province, Hong Kong and Macao Region on Marine Bioresource Conservation and Exploitation, College of Marine Sciences, South China Agricultural University, Guangzhou 510642, China

E-mail: zhangxiaoyong@scau.edu.cn

Reviewer 3 Report

Thank you very for your work ; you addressed most of my comments.

Please consider these final remarks below.

T in T-glucopyranosyl and others should be explained (terminal)

Phylogeny is still based on one strain only. Since errors may exist in databases, you should begin by several Aspergillus strains and endin by citing the most similar

The material and methods section is fine.

Line 167 : UPLC means Ultra Performance Liquid Chromatography and not ultra high pressure LC.

I am impressed by the number of flasks you used ; this could lead to important variations. You should consider higher volume flasks or bioreactors for future productions.

But you should add s to flasks in lines 88-93 if you don’t want to write the number of flasks. Because 120 mL and 25 L volumes are confusing.

Did you add information on why absorbance at 490 nm is used in Figure 1? It should be addes in 2 .2 Purification for example.

Figure 2 has been improved but still not all is readable.

In metabolomics, you should also suppress « aspartic acid and isoleucine were significantly decreased to 0.42-fold and 0.34-fold » because foldchange should be higher than 2 ; you confirmed this a few lines above in your reply letter.

Author Response

Dear reviewer (3),

Journal of Fungi,

On behalf of my co-authors, I would like to submit the second revised manuscript entitled “A new exopolysaccharide of marine coral-associated Aspergillus sp. SCAU265: Structural characterization and immunomodulatory activity” (jof-2615601 to Journal of Fungi.

The authors are grateful to the reviewers for their valuable comments and time, and all comments are responded carefully point by point.

**********************************

reviewer 3#

Thank you very for your work; you addressed most of my comments.         

Please consider these final remarks below.

  1. T in T-glucopyranosyl and others should be explained (terminal)

Response:  The first T in T-glucopyranosyl had been explained as terminal, and then others were used with the abbreviated T. Please check the second revised manuscript.

  1. Phylogeny is still based on one strain only. Since errors may exist in databases, you should begin by several Aspergillus strains and end in by citing the most similar.

Response: You are right, many errors may exist in databases (Standard databases). So the ITS sequence of the strain had been BLAST in the databases (the ITS sequences from Fungi type and reference material). Which exhibited the highest similarity of 99.62% with the type strain “Aspergillus pseudoglaucus NRRL 40 (accession number: NR135336.1)”. And the other strains were selected for construction of a phylogenetic tree. Please check the following figure.

The relevant content has been modified as follows.

Morphological characteristics and genetic analysis based on internal transcribed spacer (ITS) rRNA sequences (accession number OR122480 in GenBank) were used to compare the strain with Aspergillus pseudoglaucus NRRL 40 (accession number NR135336.1) in GenBank (Fungi type and reference material), revealing a high similarity of 99.62%. Based on this analysis, the strain SCAU265 was identified as Aspergillus pseudoglaucus [17].

  1. The material and methods section is fine.

Response: Thank you for your recognition. Your thoughtfulness and suggestion have greatly improved our manuscript.

  1. Line 167 : UPLC means Ultra Performance Liquid Chromatography and not ultra high pressure LC.

Response: Changed "ultrahigh pressure liquid chromatography" to "ultra ferformance liquid chromatography".

  1. I am impressed by the number of flasks you used ; this could lead to important variations. You should consider higher volume flasks or bioreactors for future productions.

Response: You're right. We're very sorry for our negligence. A higher volume flask (1L) with 300 mL medium was used for the fermentation. A total of 84 flasks were used and 25.2 L of fermentation broths were obtained in this study. These sentences were modified as follows.

The fungal strain SCAU265 was cultivated in a 1000 mL Erlenmeyer flask containing 300 mL of medium.

After a 7-day incubation period, the entire fermentation broth (25.2 L, a total of 84 Erlenmeyer flasks were used for the fermentation) was filtered through cheesecloth and Buchner funnel (with filter paper, pore size 30-50 μm) to remove the fungal mycelia.

  1. But you should adds to flasks in lines 88-93 if you don’t want to write the number of flasks. Because 120 mL and 25 L volumes are confusing.

Response: You're right. We're very sorry for our negligence. A higher volume flask (1L) with 300 mL medium was used for the fermentation. A total of 84 flasks were used and 25.2 L of fermentation broths were obtained in this study. These sentences were modified as follows.

The fungal strain SCAU265 was cultivated in a 1000 mL Erlenmeyer flask containing 300 mL of medium.

After a 7-day incubation period, the entire fermentation broth (25.2 L, a total of 84 Erlenmeyer flasks were used for the fermentation) was filtered through cheesecloth and Buchner funnel (with filter paper, pore size 30-50 μm) to remove the fungal mycelia.

  1. Did you add information on why absorbance at 490 nm is used in Figure 1? It should be addes in 2 .2 Purification for example.

Response: The following sentences were added into in the Part of 2.2.

The carbohydrates were first hydrolyzed into monosaccharides under the action of sulfuric acid, and rapidly dehydrated to form aldehyde derivatives. Then, they react with phenol to form orange yellow compounds, which had a maximum absorption peak at a wavelength of 490nm [21].

  1. Figure 2 has been improved but still not all is readable.

Response: I am not very sure which image is not readable. It may be due to some images being compressed in the manuscript and not having enough clarity. Now let's also upload the original images to see if there are any issues? Please check the original images (attachment, Figure 2).

  1. In metabolomics, you should also suppress « aspartic acid and isoleucine were significantly decreased to 0.42-fold and 0.34-fold » because foldchange should be higher than 2 ; you confirmed this a few lines above in your reply letter.

Response: We're very sorry for our negligence. These sentences had been removed or modified in the second revised manuscript, which were as follows.

The most prominently affected amino acid was citrulline, increased by 86.22-fold.

Among the downregulated amino acids, aspartic acid and isoleucine did not exhibite an obvious decrease (Table s1, supplemental materials), but they displayed a significant difference in amino acid metabolism (Table s2, supplemental materials).

**********************************

All revised contents in text were highlighted in red in the revised manuscript.

Thank you very much for your consideration of our manuscript for potential publication. We look forward to hearing from you soon.

Best Regards.

Sincerely yours,

Dr. Xiaoyong Zhang

University Joint Laboratory of Guangdong Province, Hong Kong and Macao Region on Marine Bioresource Conservation and Exploitation, College of Marine Sciences, South China Agricultural University, Guangzhou 510642, China

E-mail: zhangxiaoyong@scau.edu.cn
